# Sugar-sweetened beverage intakes among adults between 1990 and 2018 in 185 countries

Laura Lara-Castor [1] ✉, Renata Micha[1,2], Frederick Cudhea[1], Victoria Miller [1,3,4], Peilin Shi[1], Jianyi Zhang[1], Julia R. Sharib[1], Josh Erndt-Marino[1], Sean B. Cash [1], Dariush Mozaffarian [1,5,6] ✉ & Global Dietary Database*

Sugar-sweetened beverages (SSBs) are associated with cardiometabolic diseases and social inequities. For most nations, recent estimates and trends of intake are not available; nor variation by education or urbanicity. We investigated SSB intakes among adults between 1990 and 2018 in 185 countries, stratified subnationally by age, sex, education, and rural/urban residence, using data from the Global Dietary Database. In 2018, mean global SSB intake was 2.7 (8 oz = 248 grams) servings/week (95% UI 2.5-2.9) (range: 0.7 (0.5-1.1) in South Asia to 7.8 (7.1-8.6) in Latin America/Caribbean). Intakes were higher in male vs. female, younger vs. older, more vs. less educated, and urban vs. rural adults. Variations by education and urbanicity were largest in Sub-Saharan Africa. Between 1990 and 2018, SSB intakes increased by +0.37 (+0.29, +0.47), with the largest increase in Sub-Saharan Africa. These findings inform intervention, surveillance, and policy actions worldwide, highlighting the growing problem of SSBs for public health in Sub-Saharan Africa.

What people eat and drink is one of the most important determinants of health as well as health equity[1]. In 2019, poor dietary habits, overweight/obesity, and undernutrition contributed to 14%, 10%, and 5% of deaths in females, respectively; and 15%, 8%, and 5% of deaths in males[2]. Most of these diet-related health burdens were due to cardiometabolic diseases, including cardiovascular disease (CVD), type 2 diabetes (T2D), and cancer. These diet-related conditions not only reduce quality of life and productivity but are also leading risk factors for worse clinical outcomes from COVID-19[3]. Therefore, improving diets to reduce deaths, complications, and economic and equity burdens from cardiometabolic diseases should be a global priority.

Sugar-sweetened beverages (SSBs) are a priority concern given their relationship with obesity, CVD, T2D, cancer, and dental caries[4–6]; absence of any offsetting nutritional benefits[7]; intensive marketing to traditionally marginalized populations[8]; and contribution to health disparities[9]. Yet, for the vast majority of nations, recent national

estimates of SSB intake are unavailable, preventing an analysis of the evolving (and potentially nonlinear) trends in intake in recent decades. Furthermore, prior global estimates have not evaluated SSB intake subnationally by important sociodemographic factors such as education or urbanicity. This lack of evidence limits the ability to design and measure the impact of interventions aimed at decreasing SSB intake, as well as the capacity to identify populations where such interventions are needed the most.

In an investigation based on 2010 data, 184,000 global deaths were estimated to be attributable to SSB intake, including 72.3% from T2D, 24.2% from CVD, and 3.5% from obesity-related cancers[4]. Since that time, national and subnational dietary data from a handful of high-income nations suggest that SSB intake has decreased in selected high-income Western countries[10–15] and Brazil[16]. However, energy contribution from SSBs remains high in these nations; and SSB intake may be increasing, based on limited national intake and sales reports,

[1]Friedman School of Nutrition Science and Policy, Tufts University, Boston, MA, USA. [2]University of Thessaly, Volos, Greece. [3]Department of Medicine, McMaster University, Hamilton, Canada. [4]Population Health Research Institute, Hamilton, Canada. [5]Tufts University School of Medicine, Boston, MA, USA. [6]Department of Medicine, Tufts Medical Center, Boston, MA, USA. *A list of authors and their affiliations appears at the end of the paper. ✉e-mail: laura.lara_castor@tufts.edu; dariush.mozaffarian@tufts.edu

in non-Western countries such as Korea[17] and India[18]. More recent, harmonized, and subnationally stratified data on SSB intakes are needed to inform intervention, surveillance, and policy actions. To address these knowledge gaps, we investigated SSB intakes in adults aged 20+ years in 1990, 2005, and 2018 in 185 countries, jointly stratified subnationally by age, sex, education, and area of residence, based on new data from the Global Dietary Database (GDD).

In this work, we show that intakes were higher in male vs. female, younger vs. older, more vs. less educated, and urban vs. rural adults, and that variations by education and urbanicity were largest in Sub-Saharan Africa. SSB intakes increased between 1990 and 2018 by +0.37 servings/week (+0.29, +0.47), with the largest increase in Sub-Saharan Africa, and an overall decrease in Latin America/Caribbean. These new findings on global SSB intakes, trends, and inequities inform intervention, surveillance, and policy actions worldwide, highlighting the growing problem of SSBs for public health in Sub-Saharan Africa.

## Results

### Global, regional, and national SSB intakes in 2018

In 2018, the mean global intake of SSBs among adults was 2.7 8 oz (248 grams) servings/week (95% UI 2.5, 2.9), varying across world regions by 10-fold, from 0.7 (0.5, 1.1) in South Asia to 7.8 (7.1, 8.6) in Latin America/ Caribbean (Table 1). Among the 25 most populous countries worldwide, the highest intakes were in Mexico (8.9 [8.2, 9.8]), followed by Ethiopia (7.1 [5.6, 8.9]), the United States (4.9 [4.6, 5.1]), and Nigeria (4.9 [3.2, 7.2]), while the lowest intakes were in India, China, and Bangladesh (0.2 servings/week each) (Fig. 1, Supplementary Data 2). Of 185 countries, 58 (31.4%), representing 446 million adults or 8.9% of the adult world population, had a mean SSB intake of 7+ servings/week.

### SSB intakes by sex and age in 2018

Globally, regionally, and nationally, males had modestly higher energy-adjusted SSB intake than females (Table 1, Supplementary Table 6). The largest sex differences by region were in High-Income Countries and Latin America/Caribbean, where males consumed 0.5+ servings/week more than females across all strata (Supplementary Table 7). For example, in Latin America/Caribbean, males consumed 8.2 servings/week; and females, 7.4 servings/week. Among the 25 most populous countries, the largest differences in SSB intake between males and females were in the US, Mexico, and France, where males had 1+ serving/week higher intake overall and stratified by age, education, and urbanicity (Supplementary Table 8).

By age, SSB intakes were higher at younger vs. older ages in all regions, though with varying absolute magnitudes of intakes and differences by region (Table 1; Fig. 2). For instance, in Latin America/

**Table 1 | Global and regional mean (95% UI) sugar-sweetened beverage intakes (8 oz serving/week) in adults (20+ years) by age, sex, education, and area of residence across 185 countries in 2018**

| | World | Central/Eastern Europe and Central Asia[a] | High-Income Countries | Latin America/ Caribbean | Middle East/ North Africa | South Asia[a] | Southeast and East Asia | Sub-Saharan Africa |
|---|---|---|---|---|---|---|---|---|
| Overall | 2.7 (2.5–2.9) | 2.2 (1.9–2.5) | 3.7 (3.5–3.9) | 7.8 (7.1–8.6) | 4.6 (3.9–5.4) | 0.7 (0.5–1.1) | 0.9 (0.8–1.1) | 6.6 (5.3–8.3) |
| Sex | | | | | | | | |
| Female | 2.6 (2.4–2.8) | 2.0 (1.7–2.3) | 3.3 (3.2–3.5) | 7.4 (6.7–8.3) | 4.5 (3.8–5.4) | 0.7 (0.5–1.1) | 0.9 (0.8–1.0) | 6.5 (5.2–8.1) |
| Male | 2.8 (2.6–3.0) | 2.4 (2.1–2.8) | 4.1 (3.9–4.3) | 8.2 (7.4–9.1) | 4.6 (3.9–5.5) | 0.7 (0.5–1.1) | 0.9 (0.8–1.1) | 6.8 (5.4–8.6) |
| Age | | | | | | | | |
| 20–24 | 4.4 (4.0–4.7) | 4.6 (4.0–5.4) | 6.7 (6.4–7.0) | 11.1 (10.1–12.1) | 7.6 (6.5–9.0) | 1.0 (0.7–1.5) | 2.1 (1.9–2.5) | 7.1 (5.7–8.9) |
| 25–29 | 3.8 (3.5–4.2) | 3.7 (3.1–4.3) | 6.2 (5.9–6.5) | 10.2 (9.3–11.3) | 6.1 (5.1–7.3) | 0.9 (0.6–1.4) | 1.5 (1.3–1.8) | 7.4 (5.8–9.2) |
| 30–34 | 3.3 (3.0–3.6) | 2.9 (2.5–3.5) | 5.5 (5.2–5.8) | 9.1 (8.2–10.2) | 4.9 (4.1–5.9) | 0.8 (0.5–1.2) | 1.1 (1.0–1.3) | 7.3 (5.8–9.4) |
| 35–39 | 3.0 (2.8–3.3) | 2.4 (2.0–2.8) | 4.8 (4.5–5.0) | 8.2 (7.4–9.2) | 4.2 (3.4–5.0) | 0.7 (0.5–1.1) | 1.0 (0.9–1.2) | 7.0 (5.5–9.1) |
| 40–44 | 2.6 (2.4–2.8) | 2.0 (1.7–2.4) | 4.1 (3.9–4.3) | 7.5 (6.7–8.4) | 3.7 (3.1–4.5) | 0.7 (0.4–1.1) | 0.8 (0.7–1.0) | 6.6 (5.1–8.7) |
| 45–49 | 2.2 (2.0–2.4) | 1.8 (1.5–2.1) | 3.5 (3.4–3.7) | 6.9 (6.2–7.7) | 3.4 (2.8–4.1) | 0.6 (0.4–1.0) | 0.6 (0.5–0.7) | 6.2 (4.7–8.1) |
| 50–54 | 1.9 (1.8–2.1) | 1.6 (1.4–1.9) | 3.1 (2.9–3.2) | 6.3 (5.6–7.1) | 3.2 (2.7–3.8) | 0.6 (0.4–0.9) | 0.5 (0.5–0.6) | 5.7 (4.3–7.6) |
| 55–59 | 1.8 (1.7–2.0) | 1.5 (1.2–1.7) | 2.7 (2.6–2.8) | 5.8 (5.1–6.5) | 3.1 (2.6–3.7) | 0.5 (0.4–0.8) | 0.5 (0.4–0.6) | 5.3 (4.0–7.1) |
| 60–64 | 1.6 (1.5–1.8) | 1.3 (1.1–1.6) | 2.4 (2.3–2.5) | 5.3 (4.7–6.1) | 3.0 (2.5–3.6) | 0.5 (0.3–0.8) | 0.4 (0.4–0.5) | 4.9 (3.6–6.7) |
| 65–69 | 1.5 (1.4–1.6) | 1.2 (1.0–1.5) | 2.1 (2.0–2.2) | 5.0 (4.3–5.7) | 3.0 (2.4–3.6) | 0.5 (0.3–0.7) | 0.4 (0.3–0.5) | 4.5 (3.2–6.5) |
| 70–74 | 1.4 (1.3–1.6) | 1.2 (0.9–1.4) | 1.8 (1.7–1.9) | 4.6 (4.0–5.4) | 2.9 (2.3–3.6) | 0.5 (0.3–0.8) | 0.4 (0.3–0.5) | 4.2 (2.8–6.2) |
| 75–79 | 1.3 (1.2–1.5) | 1.1 (0.9–1.4) | 1.6 (1.5–1.7) | 4.4 (3.7–5.2) | 2.9 (2.3–3.7) | 0.4 (0.3–0.8) | 0.4 (0.3–0.5) | 3.9 (2.5–6.1) |
| 80–84 | 1.2 (1.1–1.4) | 1.0 (0.8–1.3) | 1.4 (1.3–1.5) | 4.1 (3.4–5.1) | 2.8 (2.2–3.7) | 0.4 (0.2–0.7) | 0.4 (0.3–0.4) | 3.7 (2.3–6.0) |
| 85+ | 1.2 (1.1–1.3) | 0.9 (0.7–1.2) | 1.2 (1.1–1.3) | 3.9 (3.1–4.8) | 2.8 (2.1–3.7) | 0.4 (0.2–0.7) | 0.4 (0.3–0.5) | 3.7 (2.2–6.1) |
| Education years | | | | | | | | |
| 0–6 | 2.5 (2.3–2.8) | 2.2 (1.7–2.9) | 3.7 (3.4–4.0) | 7.1 (6.3–8.1) | 4.9 (4.0–6.1) | 0.6 (0.4–0.9) | 0.9 (0.7–1.1) | 5.3 (4.1–7.0) |
| >6–12 | 2.7 (2.5–2.9) | 2.3 (2.0–2.8) | 3.8 (3.6–4.0) | 8.2 (7.3–9.2) | 4.4 (3.8–5.3) | 0.6 (0.4–1.0) | 0.9 (0.8–1.0) | 9.1 (7.2–11.4) |
| >12 | 3.0 (2.8–3.2) | 2.1 (1.8–2.5) | 3.6 (3.5–3.8) | 8.4 (7.3–9.7) | 3.7 (3.2–4.3) | 2.6 (1.7–4.1) | 1.0 (0.9–1.1) | 10.0 (7.6–13.1) |
| Area of residence | | | | | | | | |
| Rural | 2.0 (1.9–2.3) | 2.4 (2.1–2.9) | 3.7 (3.5–3.8) | 7.7 (6.8–8.7) | 5.1 (4.1–6.5) | 0.3 (0.3–0.5) | 0.9 (0.8–1.1) | 5.6 (4.4–7.0) |
| Urban | 3.2 (3.0–3.5) | 2.1 (1.8–2.4) | 3.7 (3.5–3.9) | 7.8 (7.1–8.6) | 4.3 (3.7–5.0) | 1.5 (1.0–2.4) | 0.9 (0.8–1.0) | 8.2 (6.4–10.7) |

Data are the mean intakes (95% uncertainty interval) in 8 oz servings per week. All intakes are reported adjusted to 2000 kcal/d for ages 20–74 years, and 1700 kcal/d for ages 75+ years. Data are based on a Bayesian model that incorporated up to 451 individual-level dietary surveys, and additional survey-level and country-level covariates, to estimate dietary consumption levels. SSBs were defined as any beverage with added sugars and ≥50 kcal per 8 oz serving, including commercial or homemade beverages, soft drinks, energy drinks, fruit drinks, punch, lemonade, and aguas frescas. This definition excludes 100% fruit and vegetable juices, non-caloric artificially sweetened drinks, and sweetened milk. The standardized serving size used for this analysis is 8 oz serving (248 grams). Education level: "low" 0 to 6 years of education; "medium" >6 years to 12 years of education; and "high" >12 years of education. Source data are provided as Source Data file 2.
[a]In prior GDD reports, the region Central or Eastern Europe and Central Asia was referred to as Former Soviet Union, and Southeast and East Asia was referred to as Asia.
*Oz* ounces, *SSBs* sugar-sweetened beverages, *UIs* uncertainty intervals.

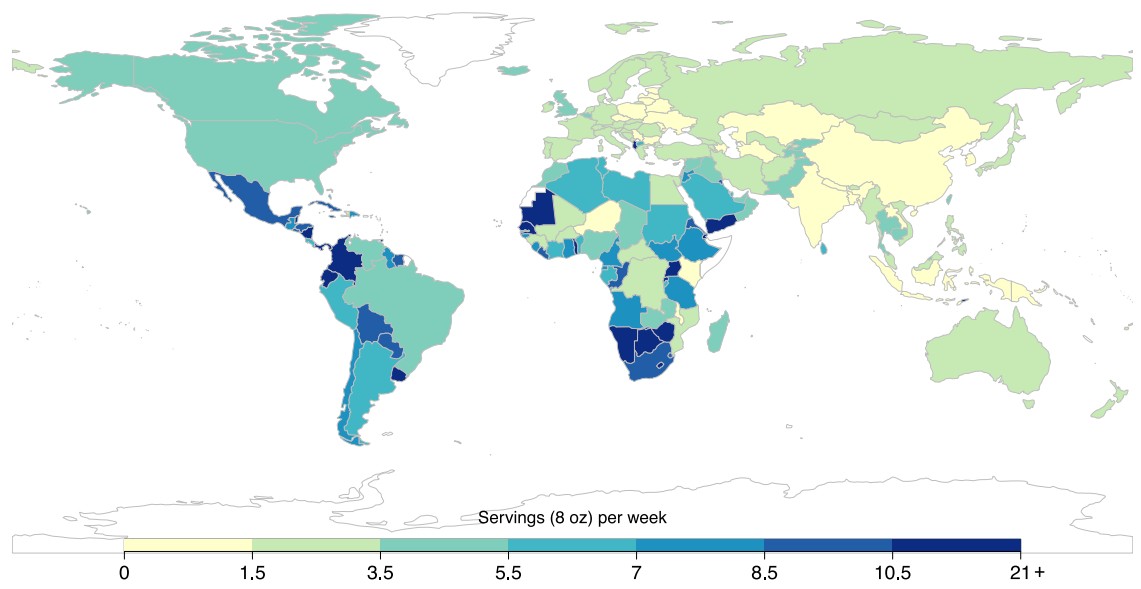

**Fig. 1 | National mean sugar-sweetened beverage intakes (8 oz servings/week) in adults (20+ years) across 185 countries in 2018.** SSBs were defined as any beverage with added sugars and ≥50 kcal per 8 oz serving, including commercial or homemade beverages, soft drinks, energy drinks, fruit drinks, punch, lemonade, and aguas frescas. This definition excludes 100% fruit and vegetable juices, non-caloric artificially sweetened drinks, and sweetened milk. The standardized serving size used for this analysis is 8 oz serving (248 grams). For this visual representation, values were truncated at 21 servings/week to better reflect the distribution of intakes globally. The analysis of the data was done using the rworldmap package (v1.3-6). Source data are provided as Source Data file 1. Oz ounces, SSBs sugar-sweetened beverages.

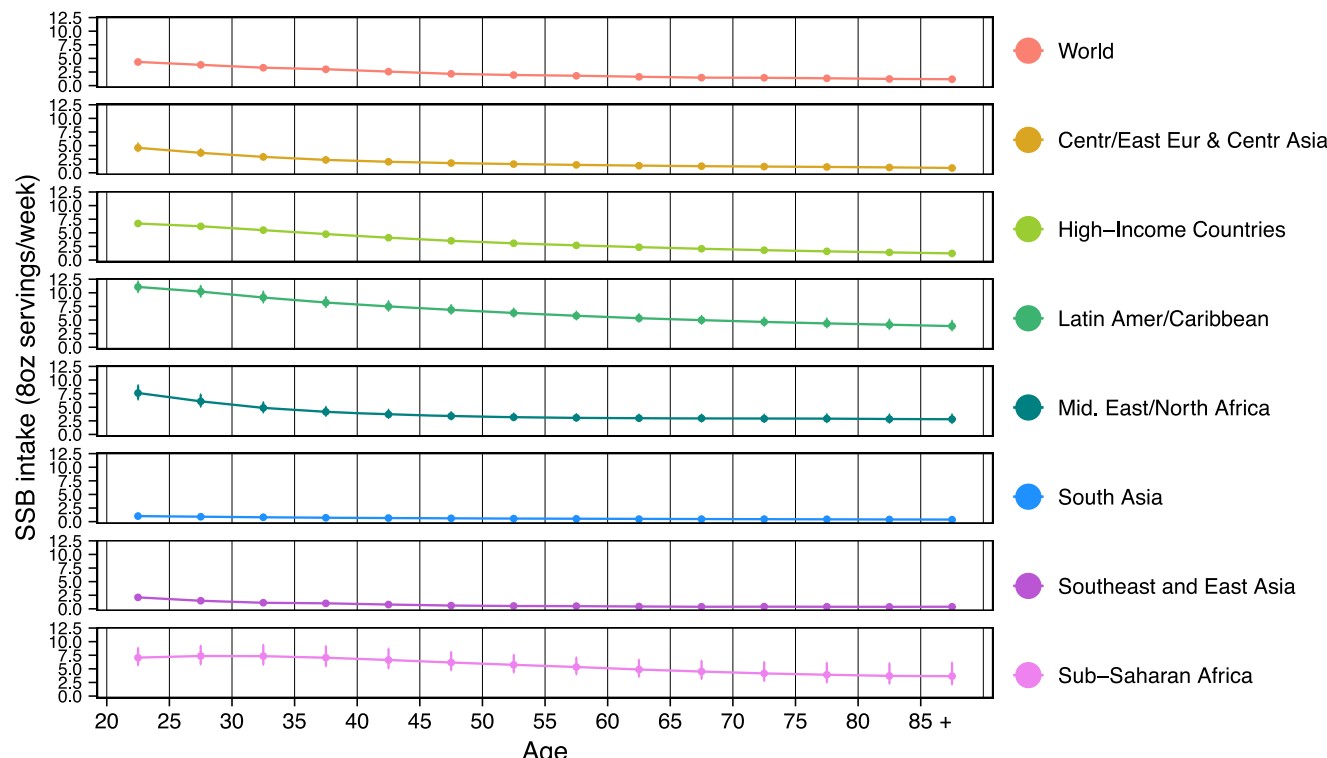

**Fig. 2 | Global and regional sugar-sweetened beverage intakes (8 oz serving/week) by age in adults (20+ years) in 2018.** SSBs were defined as any beverage with added sugars and ≥50 kcal per 8 oz serving, including commercial or home-made beverages, soft drinks, energy drinks, fruit drinks, punch, lemonade, and aguas frescas. This definition excludes 100% fruit and vegetable juices, non-caloric artificially sweetened drinks, and sweetened milk. The standardized serving size used for this analysis is 8 oz serving (248 grams). The filled circles represent the mean SSBs intake (8 oz serving/week) and the error bars of the 95% UIs. Age groups are 20–24, 25–29, 30–34, 35–39, 40–44, 45–49, 50–54, 55–59, 60–64, 65–69, 70–74, 75–79, 80–84, 85+ years. In prior GDD reports, the region Central/Eastern Europe and Central Asia was referred as Former Soviet Union, and Southeast and East Asia was referred to as Asia. Source data are provided as Source Data file 2. Centr/East Eur & Centr Asia, Central/Eastern Europe and Central Asia, GDD Global Dietary Database, Latin Amer/Caribbean Latin America/Caribbean, oz ounces, SSBs sugar-sweetened beverages, UIs uncertainty intervals.

Caribbean, where intakes were the highest compared to all regions, adults age 20–24 years had a mean SSB intake of 11.1 servings/week while adults age 85+ years had a mean intake of 3.9 servings/week. In contrast, in South Asia, where intakes were the lowest across all regions, intakes were 1.0 servings/week and 0.4 servings/week in adults age 20–24 and 85+ years respectively. Regionally, patterns of intake by age were similar between males and females (Supplementary Fig. 2). Highest weekly servings by age and region were in Latin America/Caribbean among young adults (20–24 years: 11.1; 25–29 years: 10.2), and lowest among adults age 75+ years from South Asia and 60+ years from Southeast and East Asia (-0.4 each) (Table 1). Among the 25 most populous countries, the highest intakes were among adults age 20–39 years from Mexico (11.0) and Ethiopia (7.8), and the lowest intake was among adults age 60+ years in China, India, and Bangladesh (-0.1–0.2 each) (Supplementary Table 6).

## SSB intakes by education and residence in 2018
Globally, SSB intakes varied substantially by region and education. Intakes were higher among more vs. less-educated adults in Sub-Saharan Africa (+4.68 weekly servings higher; +87.9% relative difference), South Asia (+2.00; +356.6%), and Latin America/Caribbean (+1.31; +18.3%), but lower among more vs. less-educated adults in Middle East/North Africa (−1.23; −25.2%); with smaller differences by education in other regions (Supplementary Table 7). Differences by high vs. low education decreased as age increased in all world regions, with the largest differences by education in Sub-Saharan Africa (Fig. 3a). By urbanicity, the largest differences by education were among urban and rural adults in Sub-Saharan Africa (Fig. 3b).

Among the 25 most populous countries, the largest differences in SSB intake in high vs. low-educated adults were in Pakistan, Nigeria, and Ethiopia, where more-educated adults tended to have 3+ servings/week higher intake than less-educated adults (Supplementary Table 9).

By urbanicity, global intakes were 57.3% higher in urban (3.2) vs. rural (2.0) areas (absolute difference: +1.17) (Table 1, Supplementary Table 7). By region, this pattern was most prominent in Sub-Saharan Africa (+2.67; +48.3%) and South Asia (+1.17; +340.6%); was much smaller in other regions; and was inverted in Middle East/North Africa (with lower intakes in urban regions: −0.81; −16.1%). Differences by urbanicity decreased as age increased in all regions, with largest differences in urban vs. rural adults in Sub-Saharan Africa (Fig. 4a). By region and education, differences between urban and rural areas were larger among higher educated adults in South Asia and Sub-Saharan Africa, but smaller among high educated adults in Middle East/North Africa, with little variation in other regions (Fig. 4b). Among the 25 most populous countries, largest differences by urbanicity were in Pakistan, Nigeria, and Ethiopia, where adults from urban vs. rural areas tended to have 2+ servings/week greater intake (Supplementary Table 10).

Globally, intakes were higher in urban vs. rural areas. However, regionally this was only the case in Sub-Saharan Africa and South Asia; whereas the intake was inverse in Middle East/North Africa; and intakes were almost the same between urban and rural areas in all other regions (Fig. 5a). Stratified by both education and area of residence, globally, urban adults had higher intakes than rural adults at all education levels (Fig. 5b). However, by region, this pattern was only observed in Sub-Saharan Africa and South Asia, was not notable in most other regions, and was reversed (higher intakes in rural vs. urban adults at all education levels) in Middle East/North Africa.

Strikingly, assessing both education and urbanicity by region, the highest intakes globally were among highly educated adults from urban Sub-Saharan Africa (12.4), representing 3.2% of the regional population; medium educated adults from urban Sub-Saharan Africa (11.2), representing 10.2% of the regional population; and high educated adults from urban Latin America/Caribbean (8.5), representing

15.8% of the regional population. Detailed global, regional, and national SSB intakes for 1990 and 2005 are in Supplementary Tables 11–14 and Supplementary Figs. 3–7.

## Trends in SSB intakes between 1990, 2005, and 2018
Globally, SSB intake increased from 1990 to 2018 by +0.37 servings/week ([95% UI 0.29, 0.47]; +15.9%) (Supplementary Data 2). Of note, the increase was larger from 1990 to 2005 (+0.22 [0.17, 0.28]; +9.3%) than from 2005 to 2018 (+0.15 [0.11, 0.21]; +6.0%). However, regional changes were highly heterogeneous. Between 1990 and 2005, the largest regional increases occurred in Sub-Saharan Africa (+1.04 [0.65, 1.63]; +28.6%) and High-Income Countries (+1.02 [0.94, 1.10]; +26.9%), while large decreases occurred in Latin America/Caribbean (−1.23 [−1.43, −1.03]; −14.8%) (Fig. 6, Supplementary Data 2). In contrast, from 2005 to 2018, Sub-Saharan Africa experienced the largest increase (+1.93 [1.53, 2.37]; +41.4%), while High-Income Countries experienced a significant decrease (−1.11 [−1.19, −1.03]; −23.0%). Overall, from 1990 to 2018, the largest increase was in Sub-Saharan Africa (+2.99 [2.26, 3.89]; +81.9%), while intakes rose then fell in High-Income Countries, returning close to 1990 levels by 2018. Other world regions had more modest, steady increases over time.

Energy-adjusted SSB intakes and trends were generally similar in males vs. females (Supplementary Table 15). By age, globally, and in most regions, SSB intake increased across all age groups in both time periods. However, in Latin America/Caribbean from 1990 to 2005 and in High-Income Countries from 2005 to 2018, SSB intakes decreased across all age categories. By age and region, from 1990 to 2018 SSB intakes increased most in Sub-Saharan African adults aged 20-39 (+3.28 [2.49, 4.26]; +83.3%), age 40−59 (+2.71 [1.98, 3.66]; +79.8%), and age 60+ years (+1.91 [1.28, 2.80]; +74.9%). Similarly, by education level and region, SSB intakes decreased from 1990 to 2005 in Latin America/Caribbean and from 2005 to 2018 in High-Income Countries in all education groups, while increases (or minor decreases) were seen in other regions across all education groups in both time periods (Supplementary Table 15). By education and region, from 1990 to 2018 the largest increases were in Sub-Saharan Africa among medium (+3.70 [2.69, 4.97]; +69.2%), high (+2.89 [1.95, 4.06]; +40.3%) and low (+2.72 [2.02, 3.63]; +103.6%) educated adults. Globally, SSB intakes increased from 1990 to 2018 in urban areas (+0.24 [0.16, 0.33]; +8.0%) and even more so in rural areas (+0.53 [95% UI 0.44, 0.65]; +35.3%). However, trends by urbanicity were highly heterogeneous across world regions, with larger increases in urban vs. rural areas in Central/Eastern Europe and Central Asia, and Sub-Saharan Africa; a notable decrease in urban but not rural Latin America/Caribbean, and an increase then decrease in both urban and rural areas in High-Income Countries and South Asia.

Among populous countries, the largest increases from 1990 to 2005 were in Nigeria (+4.15; +785.6%) and the US (+2.27; +42.3%); and the largest decrease in Brazil (−2.97; −40.8%) (Fig. 7, Supplementary Data 2). From 2005 to 2018, the largest increases were in Thailand (+2.09; +84.5%) and Ethiopia (+1.67; +30.8%), while the US experienced the largest decrease (−2.79; −36.5%). Overall, between 1990 to 2018, Nigeria had the largest increase (+4.33; +821.6%), followed by Thailand (+3.87; +554.9%) and Ethiopia (+3.34; +89.5%); and Brazil, the largest decrease (−2.90; −39.8%) (Supplementary Fig. 7). Trends over time by age, sex, education, and urbanicity within the 25 most populous countries are in Supplementary Discussion 1 and Supplementary Tables 15–19.

## National SSB intakes and trends according to SDI
In both 1990 and 2005, national SDI was not related to SSB intake ($r = 0.036$, $p = 0.63$; $r = −0.029$, $p = 0.69$; respectively) (Fig. 8). However, by 2018, an inverse correlation was evident between national SDI and SSB intake ($r = −0.21$, $p = 0.004$), with generally higher national intakes in countries with lower SDI.

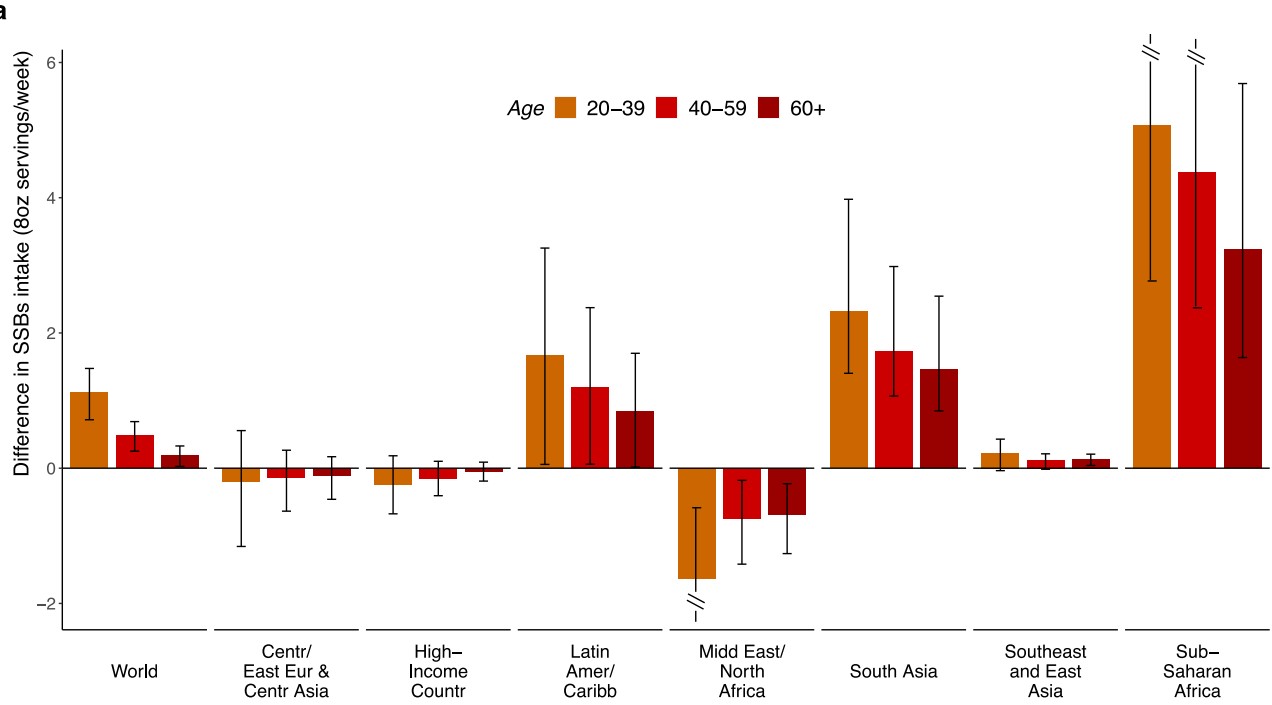

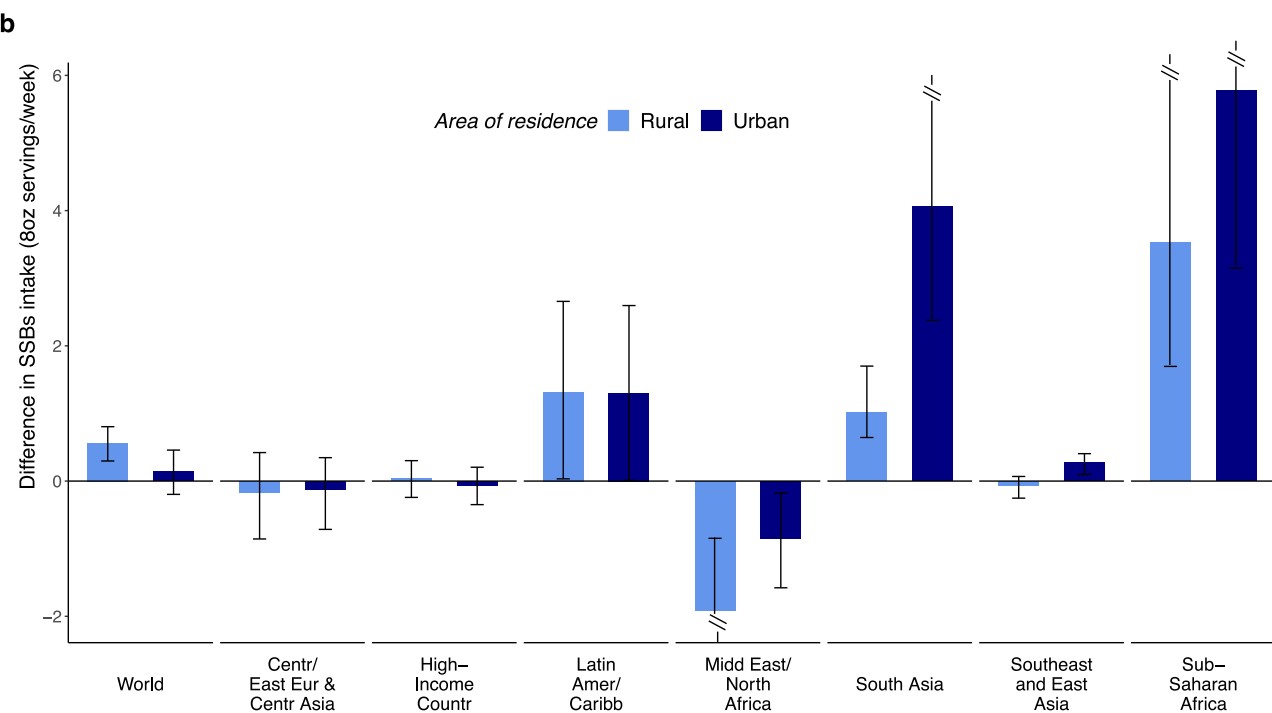

**Fig. 3 | Difference in sugar-sweetened beverage intakes (8 oz serving/week) between high vs. low educated adults (20+ years) by age and by area of residence. a** Difference SSB intakes in high vs. low educated adults by age. **b** difference in SSB intakes in high vs. low educated adults by area of residence. SSBs were defined as any beverage with added sugars and ≥50 kcal per 8 oz serving, including commercial or homemade beverages, soft drinks, energy drinks, fruit drinks, punch, lemonade, and aguas frescas. This definition excludes 100% fruit and vegetable juices, non-caloric artificially sweetened drinks, and sweetened milk. The standardized serving size used for this analysis is 8 oz serving (248 grams). The filled bars represent the mean difference in SSBs intake (8 oz serving/week) and the error bars the 95% UIs. Values were truncated at −2.0 and 5.8 (8 oz) servings/week to better represent the distribution of intakes. Upper and lower 95% UIs above or below those values are displayed with a dashed line. Colors represent the age category as "20–30 years" (orange), "40–59 years" (red), or "60+ years" (dark red); and the area of residence as "rural" (light blue) or "urban" (dark blue). In prior GDD reports, the region Central/Eastern Europe and Central Asia was referred to as Former Soviet Union, and Southeast and East Asia was referred to as Asia. Source data are provided as Source Data file 3. Centr/East Eur & Centr Asia Central/Eastern Europe and Central Asia, GDD Global Dietary Database, Latin Amer/Caribbean Latin America/Caribbean, oz ounces, SSBs sugar-sweetened beverages, UIs uncertainty intervals.

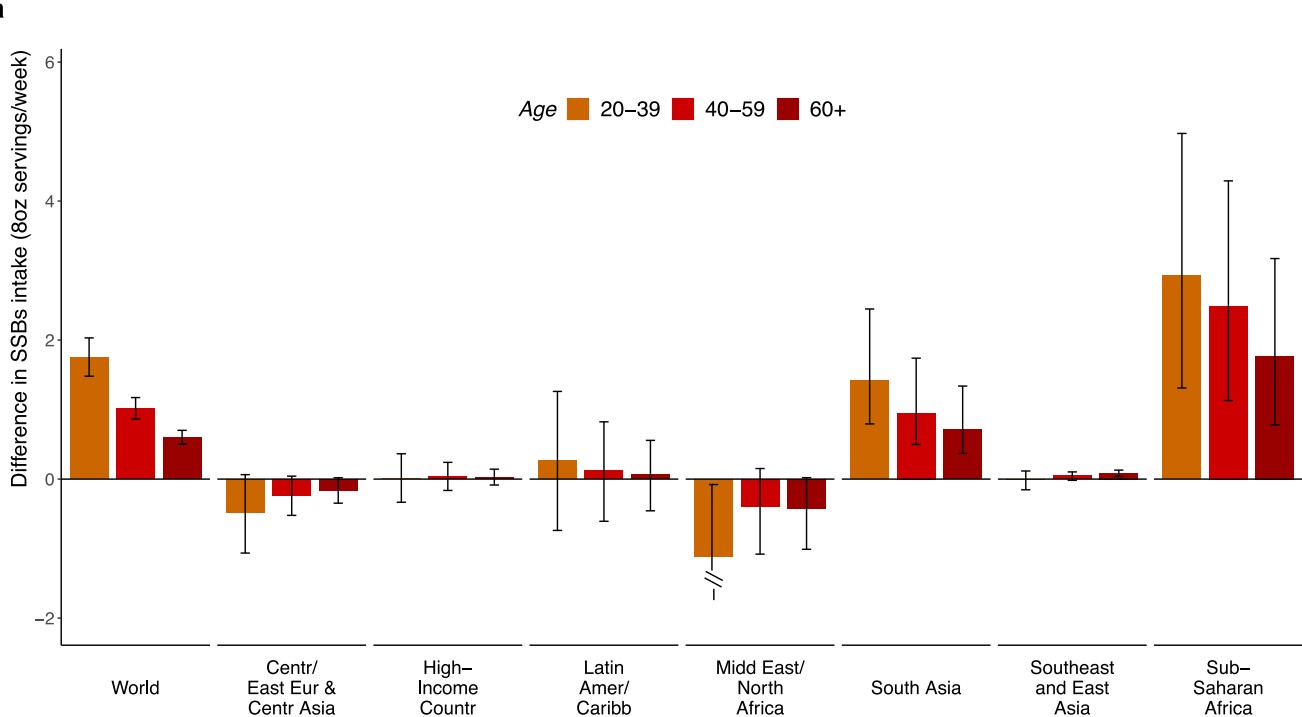

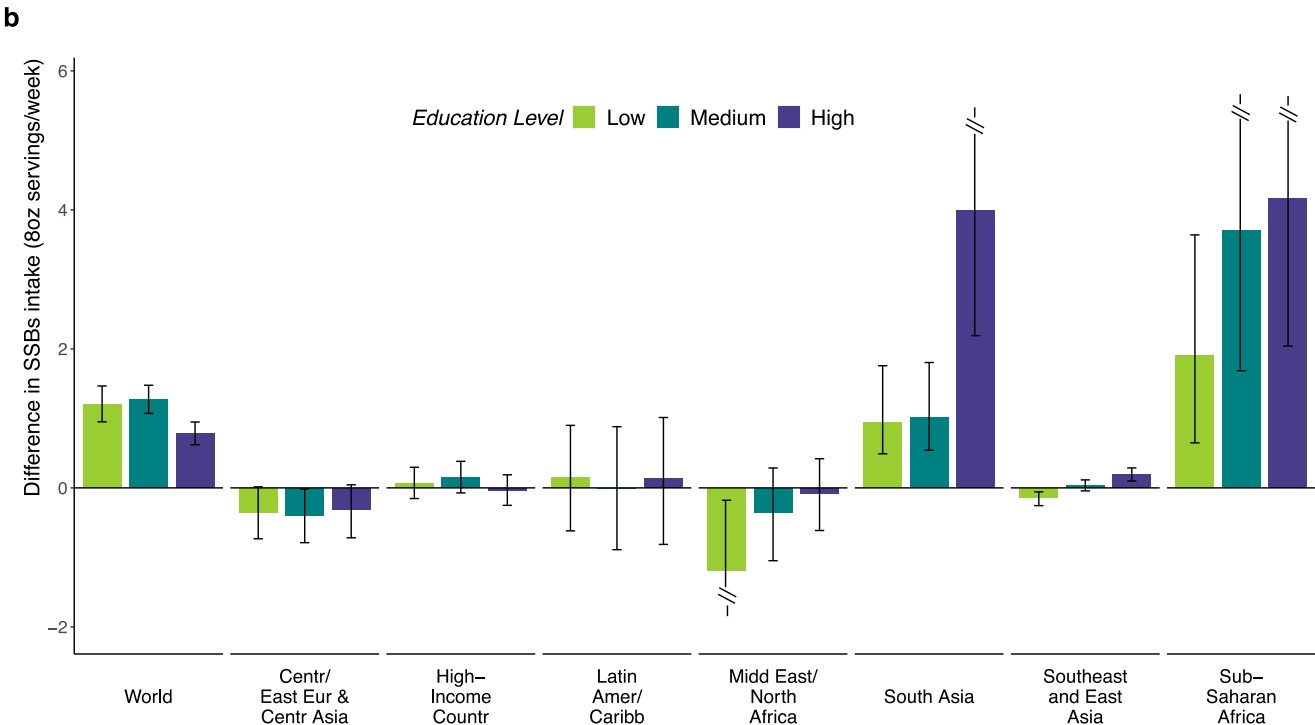

**Fig. 4 | Difference in sugar-sweetened beverage intakes (8 oz serving/week) between adults from urban vs. rural areas by age and by education level in 2018. a** Difference in SSB intakes in adults from urban vs. rural areas by age. **b** difference SSB intakes in adults from urban vs. rural areas by education. SSBs were defined as any beverage with added sugars and ≥50 kcal per 8 oz serving, including commercial or homemade beverages, soft drinks, energy drinks, fruit drinks, punch, lemonade, and aguas frescas. This definition excludes 100% fruit and vegetable juices, non-caloric artificially sweetened drinks, and sweetened milk. The standardized serving size used for this analysis is 8 oz serving (248 grams). The filled bars represent the mean difference in SSBs intake (8 oz serving/week) and the error bars the 95% UIs. Values were truncated at −2.0 and 5.8 (8 oz) servings/week to

better represent the distribution of intakes. Upper and lower 95% UIs above or below those values are displayed with a dashed line. Colors represent the age category as "20–30 years" (orange), "40–59 years" (red), or "60+ years" (dark red); and education level as "low" 0–6 years of education (light green), "medium" >6 years to 12 years of education (dark green), or "high" >12 years of education (purple). In prior GDD reports, the region Central/Eastern Europe and Central Asia was referred to as Former Soviet Union, and Southeast and East Asia was referred to as Asia. Source data are provided as Source Data file 4. Centr/East Eur & Centr Asia, Central/Eastern Europe and Central Asia; GDD Global Dietary Database; Latin Amer/Caribbean, Latin America/Caribbean, oz ounces, SSBs sugar-sweetened beverages, UIs uncertainty intervals.

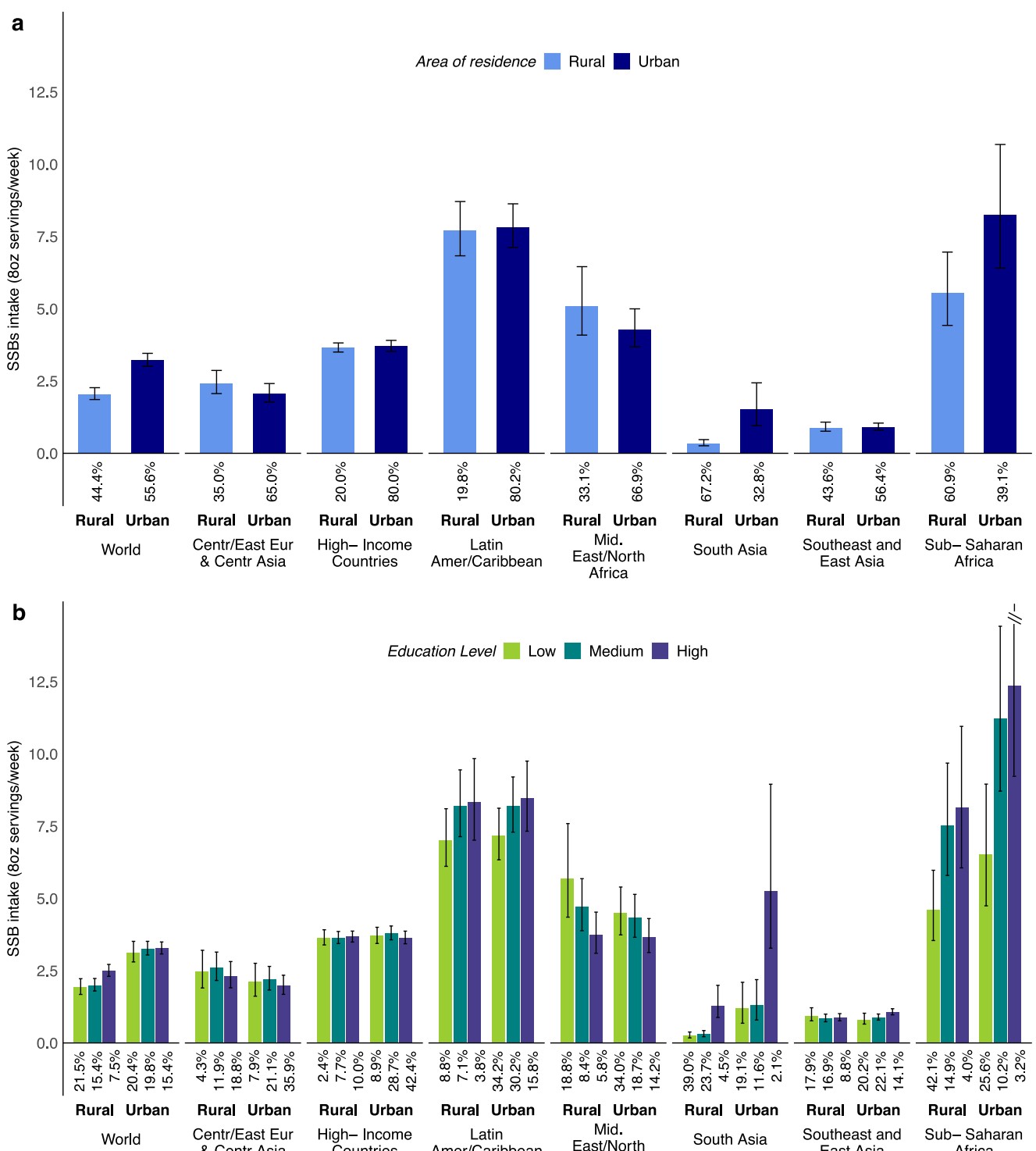

**Fig. 5 | Global and regional sugar-sweetened beverage intakes (8 oz servings/ week) in adults (+20 years) by area of residence and education level in 2018. a** SSB intakes in adults by area of residence. **b** SSB intakes in adults by area of residence and education level. SSBs were defined as any beverage with added sugars having ≥50 kcal per 8 oz serving, including commercial or homemade beverages, soft drinks, energy drinks, fruit drinks, punch, lemonade, and aguas frescas. This definition excludes 100% fruit and vegetable juices, non-caloric artificially sweetened drinks, and sweetened milk. The standardized serving size used for this analysis is 8 oz serving (248 grams). The filled bars represent the mean SSBs intake (8 oz servings/week) and the error bars the 95% UIs. Values were truncated at 14.5

(8 oz) servings/week to better reflect the distribution of intakes. Upper 95% UIs above that value are shown with a dashed line. The values below the bars correspond to the percentage (%) of the global population represented in that strata. Colors represent the education level as "low" 0 to 6 years of education (light green), "medium" >6 years to 12 years of education (dark green), or "high" >12 years of education (purple); and area of residence as "rural" (light blue) or "urban" (dark blue). In prior GDD reports, the region Central or Eastern Europe and Central Asia was referred to as Former Soviet Union, and Southeast and East Asia was referred as Asia. Source data are provided as Source Data file 2. GDD Global Dietary Database, oz ounces, SSBs sugar-sweetened beverages, UIs uncertainty intervals.

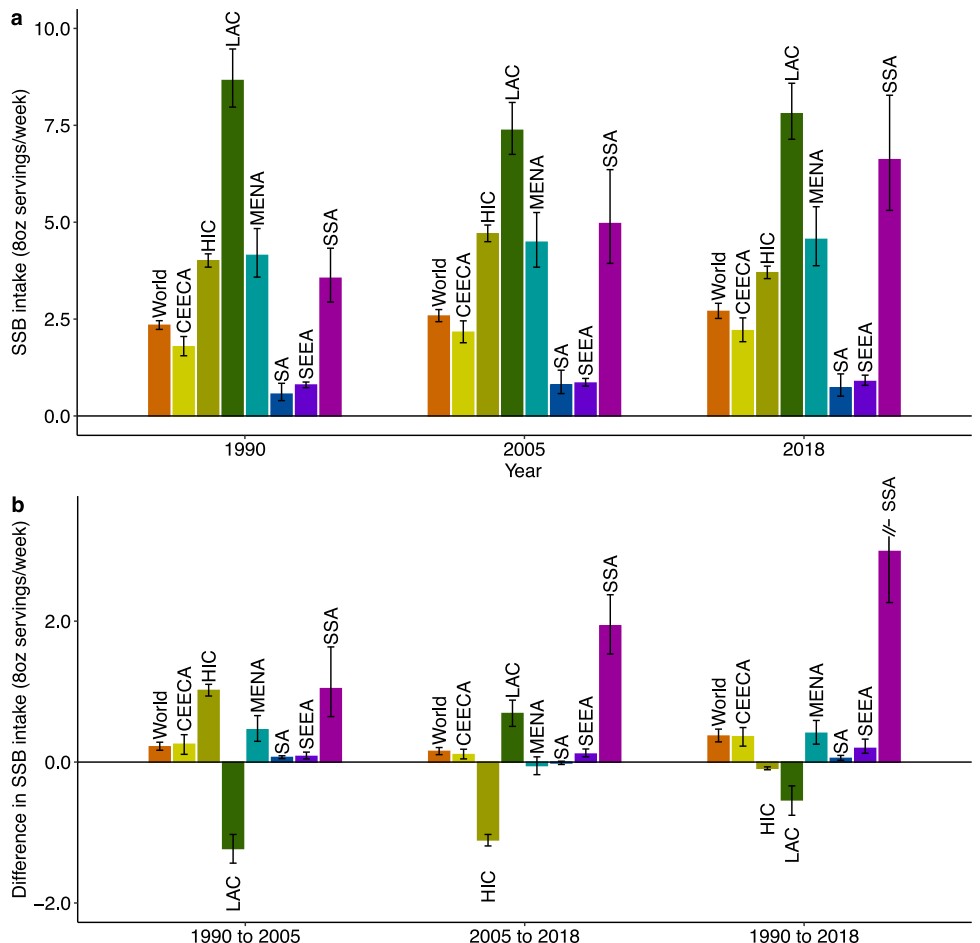

**Fig. 6 | Mean SSB intakes (8 oz servings/week) by world region in 1990, 2005, and 2018 and absolute changes from 1990–2005, 2005–2018, and 1990–2018 in adults (20+ years). a** Mean SSB intakes by world region in 1990, 2005, and 2018. **b** absolute changes in SSB intakes from 1990–2005, 2005–2018, and 1990–2018. SSBs were defined as any beverage with added sugars and ≥50 kcal per 8 oz serving, including commercial or homemade beverages, soft drinks, energy drinks, fruit drinks, punch, lemonade, and aguas frescas. This definition excludes 100% fruit and vegetable juices, non-caloric artificially sweetened drinks, and sweetened milkks. The standardized serving size used for this analysis is 8 oz serving (248 grams). The filled bars represent the mean intake (a) and the absolute change in the mean intake

(b), and the error bars represent the 95% UIs. Values in (b) were truncated at 3.5 (8 oz) servings per week to better reflect the distribution of change in intakes. Upper 95% UIs above that value are shown with a dashed line. In prior GDD reports, the region Central/Eastern Europe and Central Asia was called the Former Soviet Union, and Southeast and East Asia was called Asia. Source data are provided as Source Data files 2 and 6. GDD Global Dietary Database, oz ounces, SSBs sugar-sweetened beverages, UIs uncertainty intervals, World regions CEECA, Central/Eastern Europe and Central Asia, HIC High-Income Countries, LAC Latin America/Caribbean, MENA Middle East/North Africa, SA South Asia, SEEA Southeast and East Asia, SSA Sub-Saharan Africa.

## Discussion

This analysis, based on new GDD estimates which incorporate data on 451 mostly national, individual-level dietary surveys, provides estimates of SSB intakes and trends between 1990 and 2018 globally, regionally, and nationally. In addition to updating previous 2010 estimates stratified by sex and age with additional surveys, modeling methods, and follow-up time[19], we provide further estimates stratified by educational attainment and urbanicity. SSB intake is associated with a higher risk of obesity, CVD, T2D, cancer, and dental caries[4–6], imposing important health and economic burdens. Many national guidelines recommend limiting SSBs and keeping added sugars to <5–10% of daily calories[20–22]. Our study can help inform national dietary guidance and SSBs policies, such as SSB taxes, warning labels, marketing standards, and nutrition education, as well as the need to focus on national subgroups with inequities in SSB intake such as younger adults globally, higher educated adults in Sub-Saharan Africa, and lower educated adults from Middle East/North Africa.

Despite all efforts worldwide, from 1990 to 2018, the global intake of SSBs increased by 16%, although the magnitude of increase decelerated in 2005–2018 compared with 1990–2005. However, the more

recent global slowing, hides marked heterogeneity in regional trends, including the largest increase across all regions and time periods in Sub-Saharan Africa from 2005 to 2018 (+1.93, +41.4%) (Supplementary Data 2). Except for Latin America/Caribbean and High-Income Countries, all regions experienced steady increases in SSB intake between 1990 and 2018. Increasing trends were more pronounced in specific subnational groups, and with varying patterns in these groups by world region. For example, increases from 1990 to 2018 were larger among the youngest adults in Central/Eastern Europe and Central Asia, Middle East/North Africa, South East and East Asia, and Sub-Saharan Africa (Supplementary Table 15), while in other world regions, trends varied less by age. Trends were not notably different between males and females globally or regionally. Increases in intakes were higher in rural than in urban areas in Middle East/North Africa and Southeast and East Asia, but higher in urban than in rural areas in Sub-Saharan Africa and Central/Eastern Europe and Central Asia. By education level (a proxy of socioeconomic status), increases in SSB intakes from 1990 to 2018 were greater among the lowest-educated adults in High-Income Countries, Middle East/North Africa, and Southeast and East Asia, but greater among the highest-educated adults in South Asia and the

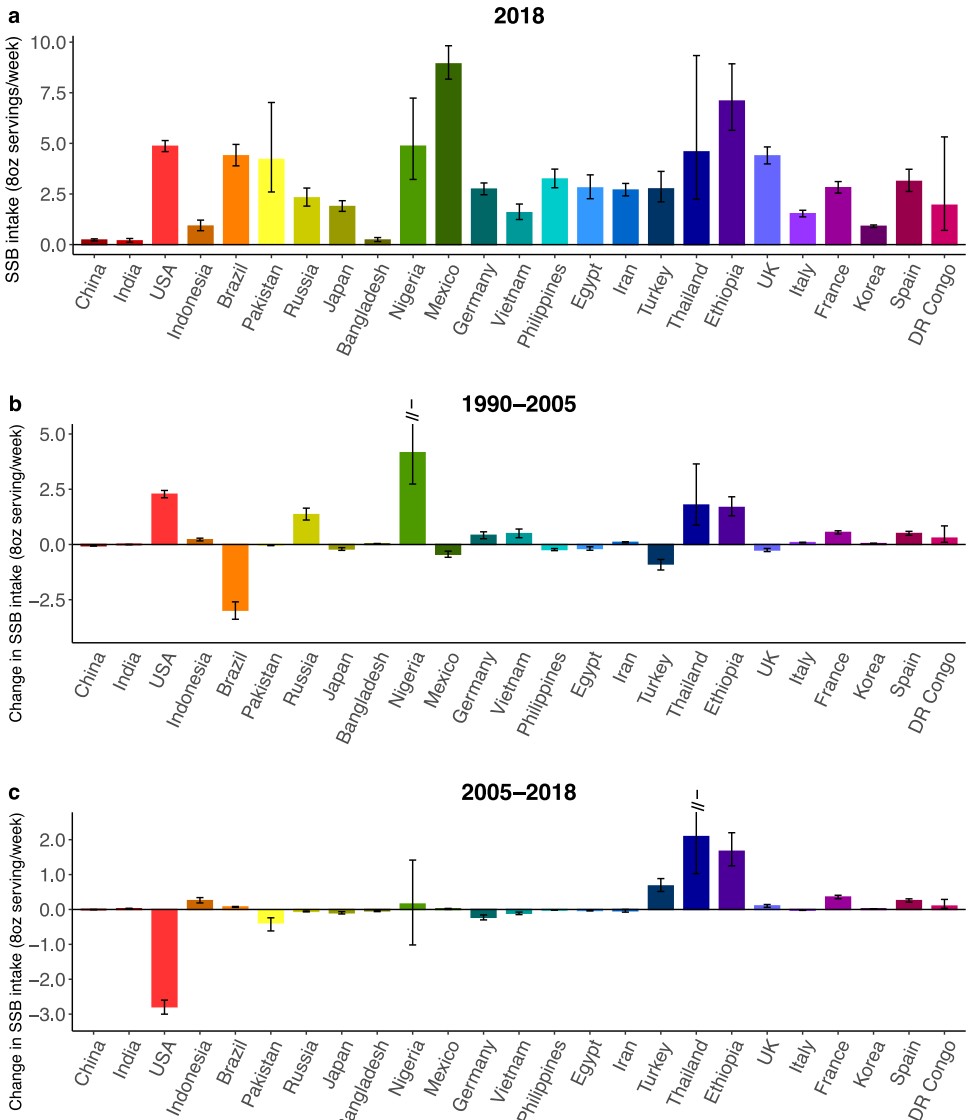

**Fig. 7 | Mean sugar-sweetened beverage intakes (8 oz servings/week) in adults (20+ years) in the 25 most populous countries in 2018, and absolute change from 1990 to 2005 and 2005 to 2018. a** Mean SSB intakes in the 25 most populous countries in 2018. **b** absolute change in SSB intakes from 1990 to 2005. **c** absolute change in SSB intakes from 2005 to 2018. SSBs were defined as any beverage with added sugars and ≥50 kcal per 8 oz serving, including commercial or homemade beverages, soft drinks, energy drinks, fruit drinks, punch, lemonade, and aguas frescas. This definition excludes 100% fruit and vegetable juices, non-caloric artificially sweetened drinks, and sweetened milk. The standardized serving size used for this analysis is 8 oz serving (248 grams). The filled bars represent the mean SSB intakes (8 oz serving/week) and the error bars the 95% UIs. Values were truncated at 5.0 (8 oz) servings/week for b and at 2.5 (8 oz) servings/week for c. Upper 95% UIs above those values are shown with a dashed line. Countries are ordered left to right from most to least populous based on 2018 adult (20+ years) population data. Source data are provided as Source Data files 1 and 7. Oz ounces, SSBs sugar-sweetened beverages, UIs uncertainty intervals.

middle-educated adults in Sub-Saharan Africa. Based on these trends, by 2018 highest SSB intakes were among the highest educated adults globally and in South Asia, Sub-Saharan Africa, and Latin America/Caribbean (Table 1).

The inverse national correlation between SDI and SSB intake in 2018 (Fig. 8) represents cross-country comparisons of national development, as opposed to within-nation socioeconomic status of individuals. The SDI findings highlight cross-national inequities in intakes, showing that higher national social and economic development is statistically significantly correlated with lower SSB intakes. In comparison, within nations, socioeconomic status as measured by education has contrasting relationships with SSB intakes in different regions, with higher educated adults often consuming more SSBs in regions mostly composed of countries with lower SDI such as in Latin America/Caribbean, South Asia, and Sub-Sharan Africa (Fig. 5). These findings are in line with the ongoing nutrition and epidemiologic transition globally, disproportionally affecting the poorest nations[23]. Moreover, they indicate a need to accelerate strategies aimed at decreasing SSB intake to tackle this global health problem, focusing on key population groups within each specific world region.

The World Health Organization (WHO) has widely recommended SSBs taxes as one of the main evidence-based policy measures to reduce intake of SSBs[24]. Nevertheless, many of these efforts have been blunted by strong food industry opposition techniques including disqualification of research findings, biased industry-funded research, misleading summaries, marketing techniques, and false claims on the potential adverse social consequences such as massive job losses[25,26]. SSB taxes have been implemented in 108 nations globally, covering 52% of the world's population, but most of these policies were implemented or updated after 2017 ($n = 71$, 66%)[27], and thus their impact is mostly not captured in SSB intakes up to 2018. Future surveillance of SSB intakes globally is needed to determine the relative impact of

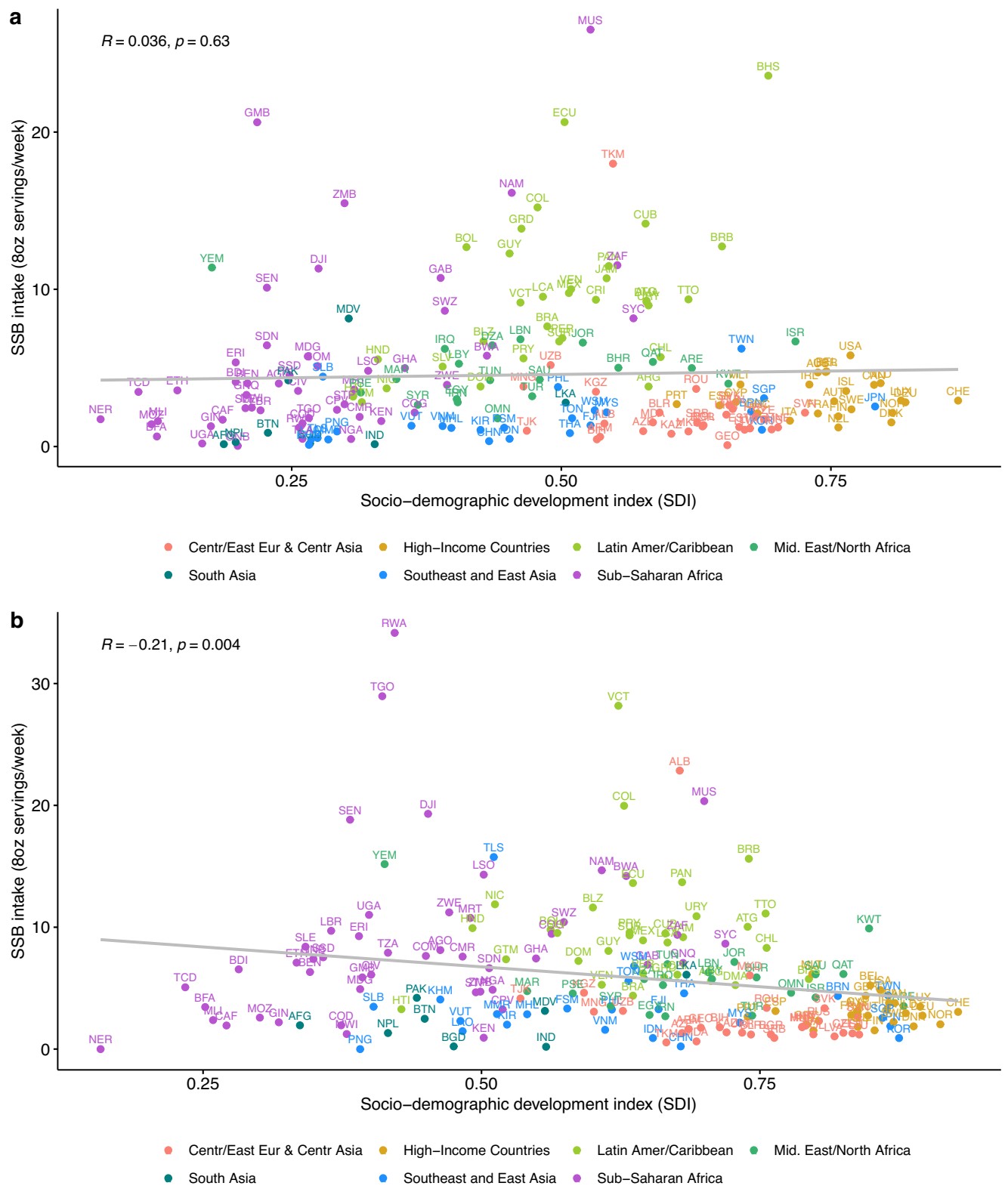

**Fig. 8 | National correlation of sugar-sweetened beverage intake (8 oz servings/week) in adults (20+ years) and sociodemographic development index by world region in 1990 and 2018 for 185 countries. a** National correlation of SSB intakes adults and SDI by world region in 1990 and **b** national correlation of SSB intakes adults and SDI by world region in 2018. Person correlation was assessed between SDI and SSB intakes among a total of 185 countries were included in this analysis. SSBs were defined as any beverage with added sugars having ≥50 kcal per 8 oz serving, including commercial or homemade beverages, soft drinks, energy drinks, fruit drinks, punch, lemonade, and aguas frescas. This definition excludes 100% fruit and vegetable juices, non-caloric artificially sweetened drinks, and sweetened milk. The standardized serving size used for this analysis is 8 oz serving (248 g). SDI was obtained from the Global Burden of Diseases study. Source data are provided as Source Data File 1. Centr/East Eur & Centr Asia Central/Eastern Europe and Central Asia, GDD Global Dietary Database, Latin Amer/Caribbean, Latin America/Caribbean, SDI sociodemographic development, SSB sugar-sweetened beverage, UI uncertainty interval.

these and other policies on SSB intakes (such as reformulation), both across nations and for subgroups within each nation.

With its economic growth and increasing middle class, Sub-Saharan Africa has become an appealing target for industry marketing of SSBs[28]. Despite evidence for the effectiveness of SSB taxes in South Africa, such measures have not been implemented in other countries in the region, limited by a lack of credible country-specific data and indicators on SSB intake to support the design, implementation, monitoring, and evaluation of taxation[29]. The findings from our study may help inform SSB-related policies in these and other countries that may be lacking valid intake estimates. In addition, we found that SSB intakes in Sub-Saharan Africa among younger, more educated, and urban adults were among the highest in the world; and that these differences by socioeconomic characteristics were also more pronounced than in other regions. These results suggest that SSB policies directed at specific subgroups could have a large impact, supporting a more strategic allocation of limited public health resources in these nations.

While Latin America/Caribbean experienced the largest decrease in SSB intake after 1990, this region also had the highest SSB intake compared to all other regions in 1990, 2005, and 2018. Our findings further suggest no major differences by education level or area of residence, emphasizing the need for implementing broad policies to decrease SSBs across the general population. In recent years, several countries in this region have implemented policies targeting SSBs, including taxes, marketing restrictions, front-of-package warning labels, and education campaigns[30]. While SSBs taxes are recommended as one of the most effective measures[24,31,32], only Chile (2014), Mexico (2014), Dominica (2015), Barbados (2015), Ecuador (2016), Peru (2018), Bermuda (2018), and Panama (2019), have implemented this policy (Supplementary Data 3)[30]. National sales analyses suggest moderate decreases in SSBs and increases in alternative beverages following implementation. In Mexico, for instance, SSB purchases decreased by 8.2% two years after tax implementation, with lower socioeconomic groups showing the largest decrease[33]. In Chile, household SSB purchases declined by 3.4% after the tax, with a higher impact among those with higher socioeconomic status[34]. Based on these successful but modest decreases, researchers and public health experts have advocated for higher tax rates and implementation across other nations to further impact SSB intakes[25].

Most national measures to reduce SSB intakes have occurred relatively recently, limiting their impact during the period of our investigation. In Latin America, for example, regulations on the advertising of SSBs have been implemented in Brazil (2006), Mexico (2009), Peru (2013), and Chile (2016); evidence on the impact of these measures remains limited, particularly for adults[30,35]. Most recently, front-of-package "black box" warning labels for certain nutrients, including added sugars, have been implemented in several countries in Latin America. In Chile, an observational study of beverage purchases found that from 2015 to 2017 the purchase volume of SSBs decreased by 22.8 ml per capita (23.7%) after implementation of the labels[36]. The findings from our study demonstrating high intakes in this region add to the evidence that additional policies are needed aimed at decreasing SSBs. Given that most national SSB policies have been recently implemented, further surveillance in this region is essential to understand the effects of these programs, especially among population subgroups with the highest intakes. Our findings provide a foundation for future time-series analyses, carefully adjusted for other factors, to assess how these and other SSB-directed policies may relate to changes in within-country SSB intake over time.

A few national studies using mostly 24-hour recalls or food records found slightly lower estimates than GDD[36,37]. However, most of these defined SSBs only as soft drinks, which generally refers only to sodas or similar beverages, thus, excluding other SSBs such as energy drinks, fruit drinks (not 100%), and homemade sweetened beverages

which were included in our estimates. Our findings might differ from estimates from sales data. However, the latter do not account for nonconsumption due to waste nor estimate a population "mean" due to the skewness of actual intakes.

We did not include sugar-sweetened milk, which are widely excluded from the definition of SSBs in policy and surveillance; nor home-sweetened teas or coffees (which often would have less than 50 kcal/serving). Home-sweetened teas and coffees were not explicitly excluded from the SSBs definition when data was requested from data owners. However, these were most likely excluded by the data owners in the majority of the cases given that tea and coffee were collected as separate variables within the same data request. Global dietary surveys generally do not collect information on sweetened tea or coffee, and even sales data on ready-to-drink tea and coffee is limited. Therefore, by excluding these products we enhance comparability across countries and findings from other reports. Furthermore, sales data suggest that ready-to-drink (RTD) tea consumption was relatively modest in Asia in 2018 (0.33 8-oz servings/week), showing slight growth in comparison to 2014[38]. Thus, the inclusion of these teas would not substantially alter our results. However, the Asian market for RTD teas, along with coffees, is expanding, and future surveillance and monitoring of the intake is needed to keep pace with this evolving category. Although fruit juices (100%) contain free sugars, observational analyses suggest no major associations of 100% fruit juice with key disease outcomes[39,40]. Therefore, these were excluded from our definition of SSBs, which focuses on beverages with added sugar.

Our definition may have missed beverages with lower sugar content, but the vast majority were captured. For instance, the calorie content per 8 oz serving is typically between 84 to 104 calories for regular sodas, 106 to 154 calories for energy drinks, 114 to 134 for juice drinks, and 114 to 216 for "aguas frescas". As a result, our definition encompasses all usual SSBs and even some with less sugar than average.

The WHO recommends limiting the intake of added sugars to less than 10% of the total caloric intake, with additional benefits at intakes lower than 5%[22]. Our findings on high intakes of SSBs and increases over time across most regions, support the global actions aimed at decreasing the intake of SSBs such as taxes. Moreover, this highlights the need to target all sugar-containing beverages, including sweetened teas and milk, to prevent substitution with these beverages.

Our study has several strengths. To our knowledge, our investigation is the first to assess and report global, regional, and national estimates of SSB intakes jointly stratified by age, sex, education, and urbanicity. Compared to previous estimates, our current model included a larger number of dietary surveys, additional demographic subgroups, and years of assessment. Updated Bayesian hierarchical models better incorporated survey and country-level covariates; assessed time trends; and addressed heterogeneity, missingness, and sampling and modeling uncertainty. Intakes were estimated from 451 surveys, mostly nationally or subnationally representative, collected at the individual-level, and representing 87.1% of the world's population. Other recent estimates of global SSBs relied mostly on national per capita estimates of food availability (e.g. FAO food balance sheets) or sales data[41]. Such findings can substantially misestimate intake compared to individual-level data[42] and are less robust for characterizing differences across population subgroups. Our estimates are informed by individual-level dietary data collected from both 24-hour recalls (24% of surveys), a gold standard for assessing nutritional intake of populations, and food frequency questionnaires (61% of surveys), a validated approach for measuring SSB intakes[43,44] (Supplementary Table 4). Finally, our findings are consistent with individual national reports[45], such as in Mexico where urban vs. rural residents and those with higher vs. lower socioeconomic status had higher SSB intakes[46]; with the opposite in the US[47], (Supplementary Table 6).

Even with systematic searches for all relevant surveys, we identified limited availability for several countries (particularly lower-income nations) and time periods[48]. Thus, estimated findings in countries with no primary individual-level surveys have higher corresponding uncertainty, informing surveillance needs to assess SSBs nationally and in subnational populations. Particularly, there were limited surveys identified for South Asia (*n* = 9) and Sub-Saharan Africa (*n* = 22), which might have limited the accuracy of our estimates in those regions (Supplementary Table 4). All types of dietary assessments include some error, whether from individual-level surveys, national food availability estimates, or other sources. Our model's incorporation of multiple types and sources of dietary assessments provides the best available estimates of global diets, as well as the uncertainty of these estimates. For instance, self-reported data relies on the memory and personal biases of the respondents, thus introducing potential bias in their responses by under or overreporting their actual intakes. Furthermore, assumptions relating to standardization of serving sizes, SSB definitions, energy adjustment, and household-level disaggregation, as well as of no interaction between sociodemographic variables in our model, could have impacted our estimates. We decided not to include interaction terms between various demographic variables to preserve model stability. To minimize these limitations, we used standardized approaches and carefully documented each survey's methods and standardization processes to maximize transparency. Overall, our findings should be taken as the best currently available, but still imperfect, estimates of SSB intake worldwide.

In conclusion, our estimates of SSB intakes reveal that the global intakes increased by 16% from 1990 to 2018, with large heterogeneity by world region and population characteristics. Our findings also provide evidence on national and subnational SSB intakes, trends over time, and related nutritional inequities, helping to inform the need and design of national and more targeted policies and approaches to reduce SSB intake worldwide, highlighting the growing problem of SSBs for public health in Sub-Saharan Africa.

## Methods

### Ethics and inclusion statement

Data informing the GDD modeling estimates for this study, including from LMICs, were collected between 1980 and 2020 in the form of dietary intake surveys. If nationally representative surveys were not available for a country, we also considered national surveys without representative sampling, followed by regional, urban, or rural surveys, and finally large local cohorts, provided that selection and measurement biases were not apparent limitations. For countries with no surveys identified, other sources of potential data were considered, including the WHO Infobase, the STEP database, and household budget survey data. As of August 2021, we identified and retrieved 1634 eligible survey years of data from public and private sources. Of these, 1225 have been checked, standardized, and approved for GDD 2018 model inclusion, of which 451 surveys inform the SSB intake estimates. Most identified data were either privately held or not in a format appropriate for our modeling. We thus relied almost entirely on direct author contacts in each country to provide us with exposure data directly. The roles and responsibilities of GDD Consortium members were determined and agreed upon before data sharing as part of a standardized data-sharing agreement.

The draft manuscript was shared with all GDD consortium members before submission for peer review, and all members have been included as authors of this work. We endorse the Nature Portfolio journals' guidance on LMIC authorship and inclusion and are committed to the inclusion of researchers from LMICs in publications from the GDD. We share the GDD data with the entire consortium, encourage authors from LMICs to take the lead on analyses and papers, and provide technical and writing support to LMIC authors. For more details on the collaborative GDD data-collection process, please visit

our website at https://www.globaldietarydatabase.org/methods/summary-methods-and-data-collection. This research is locally relevant to all countries included, given that it disaggregates findings nationally and subnationally by key demographic factors such as age, sex, education level, and urbanicity, providing decision-makers with stratum-specific SSB intake data and trends over time. This investigation was exempt from ethical review board approval because it was based on published de-identified nationally representative data, without personally identifiable information. Individual surveys underwent ethical review board approval required for the applicable local context.

### Study design

This investigation is based on a serial cross-sectional analysis of SSB intakes from the GDD 2018 for 185 countries. The GDD is an international collaborative effort to produce comprehensive and comparable estimates of dietary intakes of major foods and nutrients in 185 countries. Details on the methods and standardized data-collection protocol have been described previously and are also explained below[19,48–53]. Compared to GDD 2010, innovations include a major expansion of individual-level dietary surveys and global coverage through 2018; inclusion of updated data jointly stratified subnationally by age, sex, education level, and urban or rural residence; and updated modeling methods, covariates, and validation to improve estimates of stratum-specific mean intakes and uncertainty. This present analysis focuses on findings for adults aged 20+ years.

### Data sources

The approach and results of our survey search strategy by dietary factor, time, and region have been detailed elsewhere[48]. Briefly, we performed systematic online searches for individual-level dietary surveys in countries globally, as well as extensive personal communications with researchers and government authorities throughout the world, inviting them to be corresponding members of the GDD. Surveys were prioritized if nationally or subnationally representative and using individual-level dietary assessments with standardized 24-hour recalls, food frequency questionnaires, or short standardized questionnaires (e.g., Demographic Health Survey questionnaires). When national or subnational individual-level surveys were not identified for a country, we searched for individual-level surveys from large cohorts, the WHO Global Infobase, and the WHO Stepwise Approach to Surveillance database. Household budget surveys were used when individual-level dietary surveys were not identified for a particular country. We excluded surveys focused on special populations (e.g., exclusively pregnant or nursing women, individuals with a specific disease) or cohorts (e.g., specific occupations or dietary patterns). The final GDD model incorporated 1225 dietary surveys representing 185 countries and 99.0% of the global population in 2018. Of these, 451 surveys reported data on SSBs, totaling 2.9 million individuals from 118 countries representing 87.1% of the global population (Supplementary Tables 4–5).

Most surveys were nationally or subnationally representative (94.2%), collected at the individual level (84.7%), and included estimates in, both urban and rural areas of residence (61.6%). The total sample size included 44.3% female and 55.7% male; 70% participants from urban and 30% participants from rural areas of residence; 16.0%, 37.6%, and 46.4% participants with low, medium, and high education respectively; and 53.2% adults (18+ years) and 46.8% children/adolescents (<18 years). Further details on survey characteristics are in Supplementary Data 1.

### Data extraction

For each survey, we extracted data using standardized methods on survey characteristics and dietary metrics, units, mean, and standard deviation of intake, by age, sex, education level, and urban or rural residence[52,53]. The sociodemographic characteristics were used as

reported by each survey, and details on whether these were self-reported or measured in any other way are unavailable. Data were assessed for extraction errors and for plausibility using standardized algorithms, and survey quality by evaluating evidence for selection bias, sample representativeness, response rate, and validity of the diet assessment method (Supplementary Methods S1). Measurement comparability across surveys was maximized by using a standardized data analysis approach including averaging all days of dietary assessment to quantify mean individual-level intakes; using harmonized dietary factor definitions and units of measure across surveys; and adjusting for total energy through the residual method and using age-specific energy intakes to reduce measurement error and account for regional differences in body size, metabolic efficiency, and physical activity. All intakes are reported adjusted to 2000 kcal/day for ages 20–74 years, and 1700 kcal/day for ages 75+ years. The adult male equivalent (AME) method was used for 15 household-level surveys (3.3% of all surveys) to convert the household data to individual data based on energy requirements for a specific age and sex (Supplementary Methods 1). SSBs were defined based on their caloric content as any beverages with added sugars (as identified by product name) and ≥50 kcal per 8 oz (236.5 grams) serving, including commercial or homemade beverages, soft drinks, energy drinks, fruit drinks, punch, lemonade, and aguas frescas. This definition excluded 100% fruit and vegetable juices, non-caloric artificially sweetened drinks, and sweetened milk. All included surveys used this definition.

## Data modeling

Our model estimates intakes for years for which we have survey data available. To incorporate and address differences in data comparability and sampling uncertainty, a Bayesian model with a nested hierarchical structure (with random effects by country, region, and globally) estimated the mean consumption level of SSBs and its statistical uncertainty for each of 264 population strata across 185 countries for 1990, 1995, 2000, 2005, 2010, 2015, and 2018[53]. Although this analysis focuses only on adults aged 20+ years, the model used all age data to generate the strata estimates.

Primary inputs were the survey-level quantitative data on SSB intakes (by country, time, age, sex, education level, and urban or rural residence); survey characteristics (dietary assessment method, type of dietary metric); and country-year-specific covariates (Supplementary Methods 2). The model included overdispersion of survey-level variance for surveys that were not nationally representative or not stratified by smaller age groups (≤10 years), sex, education level, or urbanicity. The model then estimated intakes jointly stratified by age (<1, 1–2, 3–4, 5–9, 10–14, 15–19, 20–24, 25–29, 30–34, 35–39, 40–44, 45–49, 50–54, 55–59, 60–64, 65–69, 70–74, 75–79, 80–84, 85–89, 90–94, 95+ years), sex (female, male), education (≤6 years of education, >6 years to 12 years, >12 years), and urbanicity (urban, rural). Uncertainty of each stratum-specific estimate was quantified using 4000 iterations to determine posterior distributions of mean intake jointly by country, year, and sociodemographic subgroup. We computed the median intake and the 95% uncertainty interval (UI) for each stratum as the 50th, 2.5th, and 97.5th percentiles of the 4000 draws, respectively. We evaluated multiple different estimation models, and the best model was selected using five-fold cross-validation (randomly omitting 20% of the raw survey data, run five times) and validity was further assessed by comparing predicted vs. observed intakes, excluding implausible estimates (Supplementary Table 2), and by visual assessment of global and national mean intakes using heat maps. A second Bayesian model was used to strengthen time trend estimates for dietary factors with corresponding food or nutrient availability data from FAO Food Balance Sheets[54] or the Global Expanded Nutrient Supply[55].

The model incorporated country-level intercepts and slopes from these covariates, along with their correlation estimated across countries. No time component was formally included in the model; rather, time was captured by the underlying time variation in the model covariates. This model is commonly referred to as a varying slopes model structure and leverages two-dimensional partial pooling between intercepts and slopes to regularize all parameters and minimize overfitting risk[56,57]. The final presented results are a combination of these two Bayesian models, as detailed in Supplementary Methods 3.

## Statistical analysis

Global, regional, national, and within-country population subgroup intakes of SSBs and their uncertainty were calculated as population-weighted averages using all 4000 posterior estimates for each of the 264 demographic strata in each country year. Population weights for each year were derived from the United Nations Population Division[58], supplemented with data for education and urban or rural status from Barro and Lee[59] and the United Nations[60]. For the present analysis, GDD SSB estimates were collapsed for adults aged 85+ years using the 4000 simulations corresponding to the stratum level intake data derived from the Bayesian model.

Intakes were calculated as 8 oz (248 grams) servings per week. For the original SSBs definition data was requested as servings of 8 oz = 236.5 grams (Supplementary Table 1). However, for this analysis, we used the conversion 8 oz = 248 grams consistent with what has been reported in FoodData Central by the U.S. Department of Agriculture, Agricultural Research Service as the equivalence for SSBs[61]. Absolute changes and percentage changes in consumption between 1990, 2005, and 2018 were calculated at the stratum-specific level using all 4000 posterior estimates to account for the full spectrum of uncertainty and standardized to the proportion of individuals within each stratum in 2018 to account for changes in demographics over time. Stratum-specific estimates were summed to calculate the differences in intake in males vs. females, high (>12 years) vs. low (≤6 years) education, and urban vs. rural residence, further stratified by sex, age, education, and area of residence as appropriate.

We also assessed the correlation between country-level SSB intakes and the corresponding national sociodemographic development index (SDI), and how these relationships changed over time between 1990, 2005, and 2018. SDI is a measure of country/region development, ranging from 0 to 1, based on incomes per capita, average educational attainment, and fertility rates[62]. For comparisons between groups (or over time), difference thresholds were regarded as significant if the 95% UI of the difference (or change over time) did not include zero. Given the Bayesian statistical framework of our estimates, rather than frequentist null hypothesis significance testing, no p-value should be defined for statistical significance and 95% UIs of each estimate should be considered a guide[63].

## Reporting summary

Further information on research design is available in the Nature Portfolio Reporting Summary linked to this article.

# Data availability

The individual SSB intake estimate distribution data used in this as means and uncertainty (SE) for each strata in the analysis are available freely online at the Global Dietary Database (GDD, Download 2018 Final Estimates: https://www.globaldietarydatabase.org/data-download). GDD data were utilized in agreement with the database guidelines. GDD data collapsed for 85+ and by age categories 20–39, 40–59, and 60+, as well as the absolute and relative differences by strata and by year presented in this analysis, were calculated using the 4000 simulations corresponding to the stratum level intake data derived from the Bayesian model. The derived source data are provided with this paper. The 4000 simulations files can be made available

to researchers upon request. Eligibility criteria for such requests include utilization for nonprofit purposes only, for appropriate scientific use based on a robust research plan, and by investigators from an academic institution. If you are interested in requesting access to the data, please submit the following documents: (1) proposed research plan (please download and complete the proposed research plan form: https://www.globaldietarydatabase.org/sites/default/files/manual_upload/research-proposal-template.pdf), (2) data-sharing agreement (please download this form https://www.globaldietarydatabase.org/sites/default/files/manual_upload/tufts-gdd-data-sharing-agreement.docx and complete the highlighted fields, have someone who is authorized to enter your institution into a binding legal agreement with outside institutions sign the document. Note that this agreement does not apply when protected health information or personally identifiable information are shared), (3) email items (1) and (2) to info@globaldietarydatabase.org. Please use the subject line "GDD Code Access Request". Once all documents have been received, the GDD team will be in contact with you within 2–4 weeks regarding subsequent steps. Data will be shared as.csv or.xlsx files, using a compressed format when appropriate. Population weights for each strata and year were derived from the United Nations Population Division (https://population.un.org/wpp/), supplemented with data for education and urban or rural status from Barro (DOI: 0.3386/w15902) and Lee and the United Nations (https://population.un.org/wup/Download/). Source data are provided with this paper.

## Code availability

Custom code was developed using R (Version 4.0.0) for analyzing the data including aggregation of the 4000 simulations to the desired strata categories, calculation of absolute and relative differences, and summary of mean intakes globally, regionally, and nationally jointly stratified by sociodemographic group, and data visualizations including tables and figures. Given the computational size, the data aggregation, calculation of absolute and relative differences, and summary of mean intakes were run on the Tufts University High-Performance Computing Cluster (https://it.tufts.edu/high-performance-computing), supported by the National Science Foundation (grant:2018149) under active development by Research Technology, Tufts Technology Services. The statistical code can be made available to researchers upon request. Eligibility criteria for such requests include utilization for nonprofit purposes only, for appropriate scientific use based on a robust research plan, and by investigators from an academic institution. GDD will nominate co-authors to be included in any papers generated using GDD-generated statistical code. If you are interested in requesting access to the statistical code, please submit the following documents: (1) proposed research plan (please download and complete the proposed research plan form: https://www.globaldietarydatabase.org/sites/default/files/manual_upload/research-proposal-template.pdf), (2) data-sharing agreement (please download this form https://www.globaldietarydatabase.org/sites/default/files/manual_upload/tufts-gdd-data-sharing-agreement.docx and complete the highlighted fields, have someone who is authorized to enter your institution into a binding legal agreement with outside institutions sign the document. Note that this agreement does not apply when protected health information or personally identifiable information are shared), (3) email items (1) and (2) to info@globaldietarydatabase.org. Please use the subject line "GDD Code Access Request". Once all documents have been received, the GDD team will be in contact with you within 2–4 weeks regarding subsequent steps. Data will be shared as.csv or.xlsx files, using a compressed format when appropriate.

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

## Acknowledgements

This research was supported by the Gates Foundation (grant OPP1176682 to D.M.), the American Heart Association (grant 903679 to L.L.C), and Consejo Nacional de Ciencia y Tecnología in Mexico (to L.L.C.). We acknowledge the Tufts University High-Performance Computing Cluster (https://it.tufts.edu/high-performance-computing), which was used for the research reported in this paper. This material is based upon work supported by the National Science Foundation under grant number 2018149. The computational resource is under active development by Research Technology, Tufts Technology Services. The funding agencies had no role in the design of the study; collection, management, analysis, or interpretation of the data; preparation, review, or approval of the manuscript; or decision to submit for publication.

## Author contributions

Conceptualization, L.L.C, R.M. and D.M.; methodology, L.L.C, R.M., D.M. and S.B.C.; software, F.C., P.S., J.Z., J.M., V.M. and L.L.C.; validation, L.L.C., R.M. and D.M.; formal analysis, L.L.C.; investigation, L.L.C.; data curation, F.C., P.S., J.Z., V.M., J.R.S. and L.L.C.; writing—original draft preparation, L.L.C.; writing, review and editing, L.L.C., R.M., D.M. and S.B.C.; visualization, L.L.C., R.M., D.M., V.M. and S.B.C.; supervision, D.M., R.M. and S.B.C.; project administration, D.M.; funding acquisition, D.M., R.M., L.L.C.

## Competing interests

The investigators did not receive funding from a pharmaceutical company or other agency to write this report. L.L.C reports research funding from the Gates Foundation, the American Heart Association, and Consejo Nacional de Ciencia y Tecnología in Mexico, outside of the submitted work. R.M. reports research funding from the Gates Foundation; and (ended) the U.S. National Institutes of Health, Danone, and Nestle. She also reports consulting from Development Initiatives and as IEG chair for the Global Nutrition Report, outside of the submitted work. F.C., J.Z., and P.S. report research funding from the Gates Foundation, as well as the National Institutes of Health, outside of the submitted work. V.M. reports research funding from the Canadian Institutes of Health Research and from the American Heart Association, outside the submitted work. J.R.S. reports research funding from the Gates Foundation, as well as the National Institutes of Health, Nestlé, Rockefeller Foundation, and Kaiser Permanent Fund at East Bay Community Foundation, outside of the submitted work. S.B.C. reports research funding from the U.S. National Institutes of Health, the U.S. Department of Agriculture, the Rockefeller Foundation, the U.S. Agency for International Development, and the Kaiser Permanente Fund at East Bay Community Foundation, outside the submitted work. D.M. reports research funding from the U.S. National Institutes of Health, the Gates Foundation, the Rockefeller Foundation, Vail Innovative Global Research, and the Kaiser Permanente Fund at East Bay Community Foundation; personal fees from Acasti Pharma, Barilla, Danone and Motif FoodWorks; is on the scientific advisory board for Beren Therapeutics, Brightseed, Calibrate, Elysium Health, Filtricine, HumanCo, Instacart, January Inc., Perfect Day, Tiny Organics, and (ended) Day Two, Discern Dx, and Season Health; has stock ownership in Calibrate and HumanCo; and receives chapter royalties from UpToDate, all outside the submitted work. J.E.M. declares no competing interests.

## Additional information

## Global Dietary Database

Murat Bas[7], Jemal Haidar Ali[8], Suhad Abumweis[9], Anand Krishnan[10], Puneet Misra[10], Nahla Chawkat Hwalla[11], Chandrashekar Janakiram[12], Nur Indrawaty Liputo[13], Abdulrahman Musaiger[14], Farhad Pourfarzi[15], Iftikhar Alam[16], Karin DeRidder[17], Celine Termote[18], Anjum Memon[19], Aida Turrini[20], Elisabetta Lupotto[20], Raffaela Piccinelli[20], Stefania Sette[20], Karim Anzid[21], Marieke Vossenaar[22], Paramita Mazumdar[23], Ingrid Rached[24], Alicia Rovirosa[25], María Elisa Zapata[25], Tamene Taye Asayehu[26], Francis Oduor[27], Julia Boedecker[27], Lilian Aluso[27], Johana Ortiz-Ulloa[28], J. V. Meenakshi[29], Michelle Castro[30], Giuseppe Grosso[31], Anna Waskiewicz[32], Umber S. Khan[33], Anastasia Thanopoulou[34], Reza Malekzadeh[35], Neville Calleja[36], Marga Ocke[37], Zohreh Etemad[37], Mohannad Al Nsour[38], Lydiah M. Waswa[39], Eha Nurk[40], Joanne Arsenault[41], Patricio Lopez-Jaramillo[42], Abla Mehio Sibai[43], Albertino Damasceno[44], Carukshi Arambepola[45], Carla Lopes[46], Milton Severo[46], Nuno Lunet[46], Duarte Torres[47], Heli Tapanainen[48], Jaana Lindstrom[48], Suvi Virtanen[48], Cristina Palacios[49], Eva Roos[50], Imelda Angeles Agdeppa[51], Josie Desnacido[51],

Mario Capanzana[52], Anoop Misra[53], Ilse Khouw[54], Swee Ai Ng[54], Edna Gamboa Delgado[55], Mauricio Caballero[56], Johanna Otero[57], Hae-Jeung Lee[58], Eda Koksal[59], Idris Guessous[60], Carl Lachat[61], Stefaan De Henauw[61], Ali Reza Rahbar[62], Alison Tedstone[62], Androniki Naska[62], Angie Mathee[62], Annie Ling[62], Bemnet Tedla[62], Beth Hopping[62], Brahmam Ginnela[62], Catherine Leclercq[62], Charmaine Duante[62], Christian Haerpfer[62], Christine Hotz[62], Christos Pitsavos[62], Colin Rehm[62], Coline van Oosterhout[62], Corazon Cerdena[62], Debbie Bradshaw[62], Dimitrios Trichopoulos[62], Dorothy Gauci[62], Dulitha Fernando[62], Elzbieta Sygnowska[62], Erkki Vartiainen[62], Farshad Farzadfar[62], Gabor Zajkas[62], Gillian Swan[62], Guansheng Ma[62], Gulden Pekcan[62], Hajah Masni Ibrahim[62], Harri Sinkko[62], Helene Enghardt Barbieri[62], Isabelle Sioen[62], Jannicke Myhre[62], Jean-Michel Gaspoz[62], Jillian Odenkirk[62], Kanitta Bundhamcharoen[62], Keiu Nelis[62], Khairul Zarina[62], Lajos Biro[62], Lars Johansson[62], Laufey Steingrimsdottir[62], Leanne Riley[62], Mabel Yap[62], Manami Inoue[62], Maria Szabo[62], Marja-Leena Ovaskainen[62], Meei-Shyuan Lee[62], Mei Fen Chan[62], Melanie Cowan[62], Mirnalini Kandiah[62], Ola Kally[62], Olof Jonsdottir[62], Pam Palmer[62], Peter Vollenweider[62], Philippos Orfanos[62], Renzo Asciak[62], Robert Templeton[62], Rokiah Don[62], Roseyati Yaakub[62], Rusidah Selamat[62], Safiah Yusof[62], Sameer Al-Zenki[62], Shu-Yi Hung[62], Sigrid Beer-Borst[62], Suh Wu[62], Widjaja Lukito[62], Wilbur Hadden[62], Wulf Becker[62], Xia Cao[62], Yi Ma[62], Yuen Lai[62], Zaiton Hjdaud[62], Jennifer Ali[63], Ron Gravel[63], Tina Tao[63], Jacob Lennert Veerman[64], Shashi Chiplonkar[65], Mustafa Arici[66], Le Tran Ngoan[67,68], Demosthenes Panagiotakos[69], Yanping Li[70], Antonia Trichopoulou[71], Noel Barengo[72], Anuradha Khadilkar[73], Veena Ekbote[73], Noushin Mohammadifard[74], Irina Kovalskys[75], Avula Laxmaiah[76], Harikumar Rachakulla[76], Hemalatha Rajkumar[76], Indrapal Meshram[76], Laxmaiah Avula[76], Nimmathota Arlappa[76], Rajkumar Hemalatha[76], Licia Iacoviello[77,78], Marialaura Bonaccio[77], Simona Costanzo[77], Yves Martin-Prevel[79], Katia Castetbon[80], Nattinee Jitnarin[81], Yao-Te Hsieh[82], Sonia Olivares[83], Gabriela Tejeda[84], Aida Hadziomeragic[85], Amanda de Moura Souza[86], Wen-Harn Pan[82], Inge Huybrechts[87], Alan de Brauw[88], Mourad Moursi[88], Maryam Maghroun[89], Augustin Nawidimbasba Zeba[90], Nizal Sarrafzadegan[91], Lital Keinan-Boker[92], Rebecca Goldsmith[92], Tal Shimony[92], Irmgard Jordan[93], Shivanand C. Mastiholi[94], Moses Mwangi[95], Yeri Kombe[95], Zipporah Bukania[95], Eman Alissa[96], Nasser Al-Daghri[97], Shaun Sabico[97], Martin Gulliford[98], Tshilenge S. Diba[99], Kyungwon Oh[100], Sanghui Kweon[100], Sihyun Park[100], Yoonsu Cho[101], Suad Al-Hooti[102], Chanthaly Luangphaxay[103], Daovieng Douangvichit[103], Latsamy Siengsounthone[103], Pedro Marques-Vidal[104], Constance Rybak[105], Amy Luke[106], Noppawan Piaseu[107], Nipa Rojroongwasinkul[108], Kalyana Sundram[109], Jeremy Koster[110], Donka Baykova[111], Parvin Abedi[112], Sandjaja Sandjaja[113], Fariza Fadzil[114], Noriklil Bukhary Ismail Bukhary[115], Pascal Bovet[116,117], Yu Chen[118], Norie Sawada[119], Shoichiro Tsugane[119], Lalka Rangelova[120], Stefka Petrova[120], Vesselka Duleva[120], Anna Karin Lindroos[121], Jessica Petrelius Sipinen[121], Lotta Moraeus[121], Per Bergman[121], Ward Siamusantu[122], Lucjan Szponar[123], Hsing-Yi Chang[124], Makiko Sekiyama[125], Khanh Le Nguyen Bao[126], Balakrishna Nagalla[127], Kalpagam Polasa[127], Sesikeran Boindala[127], Jalila El Ati[128], Ivonne Ramirez Silva[129], Juan Rivera Dommarco[129], Simon Barquera[129], Sonia Rodríguez-Ramírez[129], Daniel Illescas-Zarate[130], Luz Maria Sanchez-Romero[130], Nayu Ikeda[131], Sahar Zaghloul[132], Anahita Houshiar-rad[133], Fatemeh Mohammadi-Nasrabadi[133], Morteza Abdollahi[133], Khun-Aik Chuah[134], Zaleha Abdullah Mahdy[134], Alison Eldridge[135], Eric L. Ding[136], Herculina Kruger[137], Sigrun Henjum[138], Anne Fernandez[139], Milton Fabian Suarez-Ortegon[140], Nawal Al-Hamad[141], Veronika Janská[142], Reema Tayyem[143], Parvin Mirmiran[144], Roya Kelishadi[145], Eva Warensjo Lemming[146], Almut Richter[147], Gert Mensink[147], Lothar Wieler[147], Daniel Hoffman[148], Benoit Salanave[149], Cho-il Kim[150], Rebecca Kuriyan-Raj[151], Sumathi Swaminathan[151], Didier Garriguet[152], Saeed Dastgiri[153], Sirje Vaask[154], Tilakavati Karupaiah[155], Fatemeh Vida Zohoori[156], Alireza Esteghamati[157], Maryam Hashemian[157,158], Sina Noshad[157], Elizabeth Mwaniki[159], Elizabeth Yakes-Jimenez[160], Justin Chileshe[161], Sydney Mwanza[161], Lydia Lera Marques[162], Alan Martin Preston[163], Samuel Duran Aguero[164], Mariana Oleas[165], Luz Posada[166], Angelica Ochoa[167], Khadijah Shamsuddin[168], Zalilah Mohd Shariff[169], Hamid Jan Bin Jan Mohamed[170], Wan Manan[170], Anca Nicolau[171], Cornelia Tudorie[171], Bee Koon Poh[172], Pamela Abbott[173], Mohammadreza Pakseresht[174], Sangita Sharma[174], Tor Strand[175], Ute Alexy[176], Ute Nöthlings[176], Jan Carmikle[177], Ken Brown[177], Indu Waidyatilaka[178], Pulani Lanerolle[178], Ranil Jayawardena[178], Julie M. Long[179], K. Michael Hambidge[179], Nancy F. Krebs[179], Aminul Haque[180], Gudrun B. Keding[181], Liisa Korkalo[182], Maijaliisa Erkkola[182], Riitta Freese[182], Laila Eleraky[183], Wolfgang Stuetz[183], Inga Thorsdottir[184], Ingibjorg Gunnarsdottir[184], Lluis Serra-Majem[185], Foong Ming Moy[186], Simon Anderson[187], Rajesh Jeewon[188], Corina Aurelia Zugravu[189], Linda Adair[190], Shu Wen Ng[190], Sheila Skeaff[191], Dirce Marchioni[192], Regina Fisberg[192], Carol Henry[193], Getahun Ersino[193], Gordon Zello[193], Alexa Meyer[194], Ibrahim Elmadfa[194], Claudette Mitchell[195], David Balfour[195], Johanna M. Geleijnse[196], Mark Manary[197], Tatyana El-kour[198], Laetitia Nikiema[199], Masoud Mirzaei[200] & Rubina Hakeem[201]

[7]Acibadem University, Istanbul, Turkey. [8]Addis Ababa University, Addis Ababa, Ethiopia. [9]Al Ain University, Abu Dhabi, UAE. [10]All India Institute of Medical Sciences, New Delhi, India. [11]American University of Beirut, Beirut, Lebanon. [12]Amrita School of Dentistry, Eranakulum, India. [13]Andalas University, Padang, Indonesia. [14]Arab Center for Nutrition, Manama, Bahrain. [15]Ardabil University of Medical Sciences, Ardabil, Iran. [16]Bacha Khan University,

Charsadda, Pakistan. [17]Belgian Public Health Institute, Brussels, Belgium. [18]Biodiversity International, Maccarese, Italy. [19]Brighton and Sussex Medical School, Brighton, United Kingdom. [20]CREA-Alimenti e Nutrizione, Rome, Italy. [21]Cadi Ayyad University, Benguerir, Morocco. [22]Center for Studies of Sensory Impairment, Aging and Metabolism (CeSSIAM), Guatemala City, Guatemala. [23]Centre For Media Studies, New Delhi, India. [24]Centro de Atencion Nutricional Antimano (CANIA), Miami, USA. [25]Centro de Estudios sobre Nutrición Infantil (CESNI), Buenos Aires, Argentina. [26]College of Applied Sciences, Department of Food Science and Applied Nutrition, Addis Ababa Science and Technology University, Addis Ababa, Ethiopia. [27]Consultative Group on International Agricultural Research (CGIAR), Montpellier, France. [28]Cuenca University, Cuenca, Ecuador. [29]Delhi School of Economics, University of Delhi, Delhi, India. [30]Departamento de Alimentacao Escolar, Sao Paulo, Brazil. [31]Department of Biomedical and Biotechnological Sciences, University of Catnia, Catania, Italy. [32]Department of CVD Epidemiology, Prevention and Health Promotion, Institute of Cardiology, Warsaw, Poland. [33]Department of Community Health Sciences, Aga Khan University, Karachi, Pakistan. [34]Diabetes Center, 2nd Department of internal Medicine, Athens University, Athens, Greece. [35]Digestive Disease Research Institute, Tehran University of Medical Sciences, Tehran, Iran. [36]Directorate for Health Information & Research, Tarxien, Malta. [37]Dutch National Institute for Public Health and the Environment (RIVM), Bilthoven, Netherlands. [38]Eastern Mediterranean Public Health Network (EMPHNET), Amman, Jordan. [39]Egerton University, Njoro, Kenya. [40]Estonian National Institute for Health Development, Tallinn, Estonia. [41]FHI Solutions, Washington, DC, USA. [42]FOSCAL and UDES, Bucaramanga, Colombia. [43]Faculty of Health Sciences, American University of Beirut, Beirut, Lebanon. [44]Faculty of Medicine, Eduardo Mondlane University, Maputo, Mozambique. [45]Faculty of Medicine, University of Colombo, Sri Lanka, Colombo 5, Sri Lanka. [46]Faculty of Medicine/Institute of Public Health, University of Porto, Porto, Portugal. [47]Faculty of Nutrition and Food Sciences of University of Porto, Porto, Portugal. [48]Finnish Institute for Health and Welfare, Helsinki, Finland. [49]Florida International University, Miami, USA. [50]Folkhälsan Research Center, Helsinki, Finland. [51]Food and Nutrition Research Institue (DOST-FNRI), Manila, Philippines. [52]Food and Nutrition Research Institute, Department of Science and Technology, Taguig City, Philippines. [53]Fortis CDOC Center for Excellence for Diabetes, New Delhi, India. [54]FrieslandCampina, Amersfoort, The Netherlands. [55]Fundacion Cardiovascular de Colombia, Bucaramanga, Colombia. [56]Fundacion INFANT and Consejo Nacional De Investigaciones Cientificas y Tecnicas (CONICET), Ciudad Autonoma de Buenos Aires, Argentina. [57]Fundacion Oftalmologica de Santander (FOSCAL), Floridablanca, Colombia. [58]Gachon University, Seongnam-si, South Korea. [59]Gazi University, Yenimahalle/Ankara, Turkey. [60]Geneva University Hospitals, Geneva, Switzerland. [61]Ghent University, Ghent, Belgium. [62]Global Dietary Database Consortium, Boston, USA. [63]Government of Canada, Statistics Canada, Ottawa, Canada. [64]Griffith University, Gold Coast, Queensland, Australia. [65]HC Jehangir Medical Research Institute, Pune, India. [66]Hacettepe University Faculty of Medicine, Ankara, Turkey. [67]Hanoi Medical University, Hanoi, Viet Nam. [68]Institute of Research and Development, Duy Tan University, Da Nang City, Viet Nam. [69]Harokopio University, Athens (Kallithea), Greece. [70]Harvard School of Public Health, Cambridge, USA. [71]Hellenic Health Foundation and University of Athens, Athens, Greece. [72]Herbert Wetheim College of Medicine, Miami, USA. [73]Hirabai Cowasji Jehangir Medical Research Institute, Pune, India. [74]Hypertension Research Center, Cardiovascular Research Center, Isfahan University of Medical Sciences, Isfahan, Iran. [75]ICCAS (Instituto para la Cooperacion Científica en Ambiente y Salud), Buenos Aires, Argentina. [76]ICMR-National Institute of Nutrition, Hyderabad, India. [77]IRCCS Neuromed, Pozzilli, Italy. [78]University of Insubria, Varese, Italy. [79]Institut de Recherche pour le Developpement, Montepellier, France. [80]Institut de Veille Sanitaire, Bobigny, France. [81]Institute for International Investigation, NDRI-USA, New York, USA. [82]Institute of Biomedical Sciences, Academia Sinica, Taipei, Taiwan. [83]Institute of Nutrition and Food Technology (INTA), University of Chile, Santiago, Chile. [84]Institute of Nutrition in Central America and Panama (INCAP), Guatemala City, Guatemala. [85]Institute of Public Health of Federation of Bosnia and Herzegovina, Sarajevo, Bosnia and Herzegovina. [86]Institute of Studies in Public Health, Federal University of Rio de Janeiro (UFRJ), Rio de Janeiro, Brazil. [87]International Agency for Research on Cancer, Lyon, France. [88]International Food Policy Research Institute (IFPRI), Washington, DC, USA. [89]Interventional Cardiology Research Center, Cardiovascular Research Center, Isfahan University of Medical Sciences, Isfahan, Iran. [90]Intitut de Recherche en Sciences de la Sante, Bobo Dioulasso, Burkina Faso. [91]Isfahan Cardiovascular Research Center, Cardiovascular Research Center, Isfahan University of Medical Sciences, Isfahan, Iran. [92]Israel Center for Disease Control, Tel-Hashomer, Israel. [93]Justus Liebig University Giessen, Giessen, Germany. [94]KLE Academy of Higher Education and Research (Deemed-to-be-University) Jawaharlal Nehru Medical College, Belagavi, India. [95]Kenya Medical Research Institute, Nairobi, Kenya. [96]King Abdulaziz University, Jeddah, Saudi Arabia. [97]King Saud University, Riyadh, Saudi Arabia. [98]King's College London, London, United Kingdom. [99]Kinshasa School of Public Health, Kinshasa, Democratic Republic of Congo. [100]Korea Disease Control and Prevention Agency (KDCA), Cheongju-si, North Chungcheong Province, Korea. [101]Korea University, Seoul, Korea. [102]Kuwait Institute for Scientific Research, Safat, Kuwait. [103]Lao Tropical and Public Health Institute, Vientiane Capital, Lao PDR. [104]Lausanne University Hospital (CHUV) and University of Lausanne, Lausanne, Switzerland. [105]Leibniz Centre for Agricultural Landscape Research, Muncheberg, Germany. [106]Loyola University Chicago, Chicago, USA. [107]Mahidol University, Bangkok, Thailand. [108]Mahidol University, Pathom, Thailand. [109]Malaysian Palm Oil Council (MPOC), Kelana Jaya, Malaysia. [110]Max Planck Institute for Evolutionary Anthropology, Leipzig, Germany. [111]Medical Center Markovs, Sofia, Bulgaria. [112]Menopause Andropause Research Center, Ahvaz Jundishapur University of Medical Sciences, Ahvaz, Iran. [113]Ministry of Health, Jakarta, Indonesia. [114]Ministry of Health, Kuala Lumpur, Malaysia. [115]Ministry of Health, Sungai Besar, Sabak Bernam, Malaysia. [116]Ministry of Health, Victoria, Seychelles. [117]University Center for Primary Care and Public Health (Unisanté), Lausanne, Switzerland. [118]NYU School of Medicine, New York, USA. [119]National Cancer Center Institute for Cancer Control, Tokyo, Japan. [120]National Centre of Public Health and Analyses (NCPHA), Sofia, Bulgaria. [121]National Food Agency, Uppsala, Sweden. [122]National Food and Nutrition Commission, Lusaka, Zambia. [123]National Food and Nutrition Institute, Warsaw, Poland. [124]National Health Research Institutes, Zhunan township, Taiwan ROC. [125]National Institute for Environmental Studies, Health and Environmental Risk Division, Tsukuba, Japan. [126]National Institute of Nutrition, Hanoi, Viet Nam. [127]National Institute of Nutrition, Hyderabad, India. [128]National Institute of Nutrition and Food Technology & SURVEN RL, Tunis, Tunisia. [129]National Institute of Public Health (INSP), Cuernavaca, Mexico. [130]National Institute of Public Health (INSP), Mexico City, Mexico. [131]National Institutes of Biomedical Innovation, Health and Nutrition, Tokyo, Japan. [132]National Nutrition Institute, Cairo, Egypt. [133]National Nutrition and Food Technology Research Institute (NNFTRI): SBMU, Tehran, Iran. [134]National University of Malaysia (UKM), Kuala Lumpur, Malaysia. [135]Nestlé Research, Lausanne, Switzerland. [136]New England Complex Systems Institute, Cambridge, USA. [137]North-West University, Potchefstroom South Africa, Potchefstroom, South Africa. [138]Oslo Metropolitan University (OsloMet), Oslo, Norway. [139]Perdana University AND Royal College of Surgeons in Ireland, Puchong, Malaysia. [140]Pontificia Universidad Javeriana Seccional Cali, Cali, Colombia. [141]Public Authority For Food and Nutrition, Sabah Al Salem, Kuwait. [142]Public Health Authority of the Slovak Republic, Bratislava, Slovak Republic. [143]Qatar University and University of Jordan, Doha, Qatar. [144]Research Institute for Endocrine Sciences, Shahid Beheshti University of Medical Sciences, Tehran, Iran. [145]Research Institute for Primordial Prevention of NCD, Isfahan University of Medical Sciences, Isfahan, Iran. [146]Risk and Benefit Assessment Department, Swedish Food Agency, Uppsala, Sweden. [147]Robert Koch Institute, Berlin, Germany. [148]Rutgers University, New Brunswick, USA. [149]Santé publique France, the French Public Health Agency, Saint Maurice, France. [150]Seoul National University, Seoul, Korea. [151]St John's Research Institute, Bangalore, India. [152]Statistics Canada, Ottawa, Canada. [153]Tabriz University of Medical Sciences, Tabriz, Iran. [154]Tallinn University, Tallinn, Estonia. [155]Taylor's University, Selangor, Malaysia. [156]Teesside University, Middlesbrough, United Kingdom. [157]Tehran University of Medical Sciences, Tehran, Iran. [158]Utica University, Tehran, Iran. [159]The Technical University of Kenya, Nairobi, Kenya. [160]The University of New Mexico, Albuquerque, NM, USA. [161]Tropical Diseases Research Centre, Ndola, Zambia. [162]Unidad de Nutricion Publica, Macul, Chile. [163]Univ Puerto Rico-Med Sci Dept Biochemistry, San Juan, Puerto Rico. [164]Universidad San Sebastian, Providencia, Chile. [165]Universidad Tecnica

del Norte, Ibarra, Ecuador. [166]Universidad de Antioquia, Medellin, Colombia. [167]Universidad de Cuenca, Cuenca, Ecuador. [168]Universiti Kebangsaan Malaysia Medical Centre, Kuala Lumpur, Malaysia. [169]Universiti Putra Malaysia, Serdang, Malaysia. [170]Universiti Sains Malaysia, Kubang Kerian, Malaysia. [171]University Dunarea de Jos, Galati, Romania. [172]University Kebangsaan Malaysia, Kuala Lumpur, Malaysia. [173]University of Aberdeen, Aberdeen, United Kingdom. [174]University of Alberta, Edmonton, Canada. [175]University of Bergen, Bergen, Norway. [176]University of Bonn, Department of Nutrition and Food Sciences, Bonn, Germany. [177]University of California Davis, Davis, USA. [178]University of Colombo, Colombo, Sri Lanka. [179]University of Colorado School of Medicine, Aurora, USA. [180]University of Dhaka, Dhaka, Bangladesh. [181]University of Goettingen, Goettingen, Germany. [182]University of Helsinki, Department of Food and Nutrition, Helsinki, Finland. [183]University of Hohenheim, Stuttgart, Germany. [184]University of Iceland, Reykjavík, Iceland. [185]University of Las Palmas de Gran Canaria (ULPGC), Canary Islands, Las Palmas, Spain. [186]University of Malaya, Kuala Lumpur, Malaysia. [187]University of Manchester, Manchester, United Kingdom. [188]University of Mauritius, Reduit, Mauritius. [189]University of Medicine and Pharmacy Carol Davila, Bucharest, Romania. [190]University of North Carolina at Chapel Hill, Chapel Hill, USA. [191]University of Otago, Dunedin, New Zealand. [192]University of Sao Paulo, Sao Paulo, Brazil. [193]University of Saskatchewan, Saskatoon, Canada, Saskatoon, Canada. [194]University of Vienna, Vienna, Austria. [195]University of the Southern Caribbean, Port-of-Spain, Trinidad and Tobago. [196]Wageningen University, Wageningen, Netherlands. [197]Washington University in St. Louis, St. Louis, USA. [198]World Health Organization (WHO), Amman, Jordan. [199]World Health Organization (WHO), Geneva, Switzerland. [200]Yazd Cardiovascular Research Centre, Shahid Sadoughi University of Medical Sciences, Yazd, Iran. [201]Ziauddin University Karachi, Karachi City, Pakistan.

