## [Peer Review File · Nature Communications]

REVIEWER COMMENTS

Reviewer #1 (Remarks to the Author):

Overall, the study was very well-conducted, with robust methods, solid data, relevant figures, and interesting results. We applaud the tremendous effort it must have required to assemble this dataset; surely it will provide many rich insights about global nutrition, including the present study.

However, there were several concerns that need to be addressed to warrant publication. First, the sheer volume of the results was hard to absorb, and it was sometimes hard to extract the most important findings. The supplemental material was nearly twice as long as the paper; moving some results (and their accompanying appendix materials) into a separate paper could help streamline reporting of results and interpretation (for example, focusing this paper on region-level data and then a separate paper on countries or on some of the sub-population differences). In addition, the interpretation of the results (in the discussion section) could have been more in-depth. Moreover, there were a few methodological concerns that warrant additional analysis, the most serious of which relate to how SSBs were defined and better defining how data points were extrapolated to estimate trends. With regards to SSB definitions, the concern is that not using the correct definition of SSBs (most notably omitting coffees, teas, and sweetened milks) could lead to serious bias in global estimates and this would differentially impact some regions more than others. These revisions would make for a much stronger paper that would contribute substantially to the public health literature.

Major comments

- It is unclear how valid it is to use data after 2015 in countries where you didn't have actual data points after then. Predicting the trends between two time points makes sense, but extrapolating earlier trends into the future seems to defeat the purpose of the exercise, which is that trends may be changing and we want to understand them.
- More information needed about SSBs definition: did you have access to added sugar information in all surveys? Or were SSBs generally defined by category? Why was the 50kcal/8 oz serving threshold used, given that most SSB policy is on products with >0g added sugar or based on a sugar threshold (not a calories one)?
- I think the very low estimates in Asia are due to the omission of teas. In country-reports show very different estimates of SSB intake when these are included. For example, if you look up country reports by the ministry of health or NGOs (Bangladesh heart foundation) you find a very different estimate of SSBs than what is presented in this study. It is not clear why authors did not include sweetened milk, tea, and coffee in their definition of SSBs. Perhaps it is due to the complexity of defining these in the data; however every effort should be made to do this, or else the main results of this study will be

seriously biased. Many of these drinks are pre-sweetened and purchased ready to drink and should absolutely be included. Some will have sugar added during preparation or at the table; every effort should be made to include these, just as you include other homemade drinks. Even those these could not be included in policies like SSB taxes, since they don't contain added sugar at the time of purchase, they are critical for estimating consumption, since they contain added sugar at the time of consumption.

- Similarly, consider including sweetened milks. Sweetened milks are controversial since dairy contributes some benefits; yet at the same time, their exempt status from SSB taxes has largely been the product of industry lobbying. The WHO recommends including sweetened milks in SSB definitions for policy purposes (<https://www.who.int/publications-detail-redirect/9789240056299>). At a minimum, I suggest a sensitivity analysis of your main findings including coffee, tea, and sweetened milks.

- You could also consider including 100% juices. Though these do not contain added sugar, they do contain free sugars and are identified by the WHO as sugary drinks that should be limited or reduced (<https://www.who.int/publications/i/item/9789241549028>).

- FFQs and the corresponding method to convert into servings most likely underestimate the consumption, which could have biased the results towards the null, particularly among high consumers. It would be helpful to include a table showing the % of studies that used different diet assessment methods as well as other relevant information on the diet data used (average sample size & range, average calories intake & range, etc).

- Include prevalence of consumers. This result is essential. Particularly for the low-consuming countries, the servings per week are less informative. What % of people are even consuming these drinks once a week? You present data on the high consumers but I also want to know the "any" consumers.

- In general, the discussion was lacking in-depth integration and interpretation of results. Specific suggestions follow:

- The discussion section could have provided more context about differences among the regions in terms of the economy, nutrition and epidemiologic transition, availability and accessibility of SSBs, and existent policies, similar to what was done with Sub-Saharan Africa. Also, results from the implementation of policies that could have reduced SSBs intake besides taxation, particularly in Latin America, could have been included, as well as potential consequences, such as product reformulation.

- In addition to it, the discussion could have provided a comparison between the magnitude of the results and previous studies of SSBs, particularly in high-income countries, as well as age-specific policies, mostly notably Chile's marketing ban, although results clearly showed that intake was higher among younger age groups. Including an interpretation of the results in comparison to current recommendations for added sugars/SSBs would also have been helpful.

- The discussion section did not mention sex-differences in SSBs intake, and how they could have been due to energy adjustment procedures. Also, those and the adult male equivalent (AME) did not take into consideration regional differences in body composition, which could have impacted the results. These and other potential sources of bias could have been discussed with more detail.
- How do your results compare with other, potentially more objective data (i.e., Euromonitor sales data)? Self-reported dietary data would be much more compelling in this case if at least it shows similar trends to what is reported in other sources.

Minor comments

- Abstract: is it possible to include 95% CI?
- I'm not sure if reporting the mean global increase in SSB servings is that useful, because surely it disguises a great deal of heterogeneity across countries as well as consumption. Could you present averages by region? Or, change in the prevalence of SSB consumers? I think this may be more telling.
- Background: implies SSB intake causes social inequities, whereas it is the reverse that is true.
- Background: If the framing is that nutrition leads to morbidity and mortality globally, I think you need to mention undernutrition as well as overnutrition, i.e., the double burden. SSBs fit into this picture also, given that they do not contribute nutritional benefits and yet, in many countries, have been promoted/marketed to low-income populations, exacerbating the double burden.
- Methods: I would like to have seen family income included in the stratifications, mostly as another source of inequality that urbanicity and education attainment alone might not have evidenced.
- Although the methods are robust, the Bayesian method only provide an estimate and did not indicate differences between subgroups. Please further clarify in the manuscript text that you are extrapolating trends and not making estimates based on observed data points.
- Did you include interaction terms in the model (for example, interaction between sex/education or education/urbanicity, and age/urbanicity)?
- How was missing data handled? If you lacked dietary data on an age group in a country, for example, how did you include this? Consider conducting sensitivity analyses to determine if the imputation method changed results.
- It is not clear how the SDI was used- you controlled for this? It's not clear what "assessing trends according to SDI" means.
- Why use means instead of medians or quantile regression? Was the distribution of SSB intake right skewed, as is typical of dietary data?

- Results: For looking at age, I think it is less interesting to compare age groups across countries or regions than to look at the old- young age difference within each country and then compare those. You've already told us that SSB intake is low in Asia relative to Lat Am, so presenting the findings that younger adults in Asia are also low is less interesting. Do they consume more than older adults?
- Results: Useful to present urbanicity results separately before by education
- Results: Line 133- this is the first time I'm seeing you use the "high income country" categorization which seems heterogeneous to how you've presented the results so far. Also- where does Europe & Australia fit in here? High-income countries could be a useful way to distinguish global trends but be sure to define & use consistently throughout .
- Line 156: typo (Sub-Sharan)
- Supp page 52: typo (Global Fund Research Program)

Thanks for the opportunity to review this paper. I look forward to reading future versions and seeing the eventual publication.

Reviewer #2 (Remarks to the Author):

The present study provides novel evidence on the consumption of sugar-sweetened beverage consumption for 185 countries and provides novel insights into time trends as well as variation in consumption levels by education and urbanicity. The authors provide a very comprehensive explanation of the complex methodological approach used by them. The results are timely and of substantial value for policy makers worldwide.

Below are some comments that this reviewer would like to see addressed before publication of the paper:

Abstract: Compared to the wealth of data provided in the article the abstract is quite vague. This applies specifically to the conclusion which should be more specific. The authors are encouraged to shorten the background sentence and to incorporate some of the novel findings (e.g. that SSB consumption emerges as a public health problem in sub-Saharan Africa).

US-centric presentation: Given that this is a global report and that – as acknowledged by the authors – most countries report intakes in g (or ml) it is surprising that all data are still given servings based on ounces. At the very least should the authors include the g/ml equivalent in the abstract. Note that in Table S6 the conversion is explained as 8oz=237 g, whereas in most other footnotes 8oz=248 g is given.

Similarly, the authors should not only refer to “added sugar”, but also to “free sugar”, i.e. the WHO definition which is used by many countries around the globe.

Country-specific sources of data: while this reviewer acknowledges that a complex model was used to estimate the consumption in the respective countries and that this approach allowed to report intake levels for 185 countries despite the fact that at least one report of survey data was only available for 118 countries it would be very informative for both the reader and local policy makers if the authors provided an overview of the data sources used for the estimates in each country.

Differences by education vs relation of SSB intake to SDI: At first glance it appears contradictory that intake levels should be higher among persons with a higher educational level in most regions whilst on the other hand Figure 5 suggests an inverse correlation between national SDI and SSB intake for 2018. This is presumably due to the former being an observation based on individual data while the latter is a correlation on national level. Yet, this reviewer feels that this should be explained in the discussion together with the implications arising from this for policy makers.

Reference to tax policies: While it is correct that many Latin American countries still have no taxes implemented this is however the region (together with Europe) with the broadest coverage of tax policies. This could be acknowledged in the context of the discussion of the results for this region.

Minor:

- Page 15, lines 23-26: this is difficult to understand. Please, explain more explicitly what is hidden and why this in line with the inverse correlation seen in 2018.
- Please, mention in the introduction that this report focuses on findings for adults aged 20 years onwards.
- Please, explain why out of 1.248 surveys only 451 report data on SSB
- Line 139 ff, page 20: please explain that energy adjustment was made via the residual method whenever possible. This is explained in the supplemental material, yet readers not familiar with energy adjustment may interpret reference to 2000 kcal or 1700 kcal as “energy adjustment”.
- Lines 151 ff, page 21: This reviewer struggles to understand why data from children and adolescents were needed to estimate intake levels among adults. A short explanatory would be much appreciated.

Reviewer #3 (Remarks to the Author):

The paper is of high quality but I have some comments/queries as described below.

I have carried out a statistical review:

1. The methods section refers to the posterior predictive. Should this be the posterior? That would be the more natural estimate to report.
2. As part of the cross-validation, was any check made of the validity of the UIs? If not, this should be done.
3. In 'Statistical Analysis', I think that 'posterior predictions' should be replaced by 'posterior estimates' (consistent with point 1).

Other minor comments:

1. I think the metric equivalent of the 8 oz serving should be given in the findings and main text, particularly as the paper is intended for an international policymaking audience.
2. In the introduction, is it correct to say that the poor dietary habits etc definitely caused these deaths? Would text such as 'are associated with' or 'contributed to' be more accurate?
3. In Figure 2, I think joining the points is misleading unless it really is intended that readers should interpolate between them. I'd suggest removing the lines and making the points and Crls larger.
4. In the Figure 4 caption, "the filled bars represent the absolute change..." seems to refer to (B) only. Should the text be modified here?
5. A further limitation that should be added in that section is that the data are self-reported. Is there anything in the literature on how well people taking the surveys recall their SSB consumption?
6. This is more an editorial point, but should it be stated somewhere which country the Consejo Nacional de Ciencia y Tecnología is from?

The supplementary material should be proof read to the standard of the paper. There are quite a few typos, including some in mathematical formulae. Some are mentioned below.

Specific comments:

1. Typo in caption for Fig S1 ('standardize')
2. On p10, could the use of entry and exit points be clarified? It's not clear what p is here.
3. p11. typo for 'descriptions' at the bottom of the page
4. p13. typo for 'standardized' at top of page.
5. p14. various subscripts are not subscripted in the text, and in the description of the age model: 'Zh'
6. p14. I think this should be 'Estimates' rather than 'Predictions'. It looks like you're (correctly) reporting the posterior rather than the posterior predictive. I'm not sure why you wouldn't have a posterior estimate of the estimate for a country with no data as this would be informed by the other data and the covariates; similarly with superregions with no data.
7. For the varying slopes model, it would be helpful to include the time-varying component more explicitly in the mathematical representation by adding time-based subscripts. Is it that you have $\mu_{i,t} = \alpha_{\text{country}[i, \text{reference year}] + \beta_{\text{country}[i]} * \text{FAO}_t + M_{\text{method}[i]}$?

REVIEWER COMMENTS

Reviewer #1 (Remarks to the Author):

Overall, the study was very well-conducted, with robust methods, solid data, relevant figures, and interesting results. We applaud the tremendous effort it must have required to assemble this dataset; surely it will provide many rich insights about global nutrition, including the present study.

Response: Thank you for your kind and positive comments.

However, there were several concerns that need to be addressed to warrant publication. First, the sheer volume of the results was hard to absorb, and it was sometimes hard to extract the most important findings. The supplemental material was nearly twice as long as the paper; moving some results (and their accompanying appendix materials) into a separate paper could help streamline reporting of results and interpretation (for example, focusing this paper on region-level data and then a separate paper on countries or on some of the sub-population differences). In addition, the interpretation of the results (in the discussion section) could have been more in-depth. Moreover, there were a few methodological concerns that warrant additional analysis, the most serious of which relate to how SSBs were defined and better defining how data points were extrapolated to estimate trends. With regards to SSB definitions, the concern is that not using the correct definition of SSBs (most notably omitting coffees, teas, and sweetened milks) could lead to serious bias in global estimates and this would differentially impact some regions more than others. These revisions would make for a much stronger paper that would contribute substantially to the public health literature.

Response: We appreciate the consideration of our results to be substantial enough to be presented in two separate manuscripts. After discussion with the journal editors, we decided to keep the presentation as one manuscript and ensure the results have a proper flow and are as clear as possible to the reader. The methodological comments are answered below under each specific comment.

Major comments

•Comment 1: It is unclear how valid it is to use data after 2015 in countries where you didn't have actual data points after then. Predicting the trends between two time points makes sense, but extrapolating earlier trends into the future seems to defeat the purpose of the exercise, which is that trends may be changing and we want to understand them.

Response: Our model estimates intakes for years for which we have survey data available. We searched for and included dietary surveys conducted up to 2018, which is the latest year for which we report estimates (Supplementary Tables 4-5, **pg. 21-22**). Specifically, our Bayesian hierarchical modeling methods estimated SSB intakes using the 451 dietary surveys identified and, importantly, several country-specific and time-specific annual covariates up to 2018 for 264 population strata across 185 countries for 7 time points. Time was captured by the underlying time variation of FAO/Genus covariates, as well as a second Bayesian model capturing the covariate variables that change with time. This has been clarified in the Methods section.

Changes to text (pg. 20; Line 439): “Our model estimates intakes for years for which we have survey data available.”

Changes to text (pg. 21-22; Line 461-470): “A second Bayesian model was used to strengthen time trend estimates for dietary factors with corresponding food or nutrient availability data from FAO Food Balance Sheets or the Global Expanded Nutrient Supply. The model incorporated country-level intercepts and slopes from these covariates, along with their correlation estimated across countries. No time-component was formally included in the model; rather, time was captured by the underlying time variation in the model covariates. This model is commonly referred to as a varying slopes model structure and leverages two-dimensional partial pooling between intercepts and slopes to regularize all parameters and minimize overfitting risk. The final presented results are a combination of these two Bayesian models, as detailed in Supplementary Methods 3.”

• **Comment 2:** More information needed about SSBs definition: did you have access to added sugar information in all surveys? Or were SSBs generally defined by category? Why was the 50kcal/8 oz serving threshold used, given that most SSB policy is on products with >0g added sugar or based on a sugar threshold (not a calories one)?

Response: SSBs were defined based on product name and caloric content as “any beverages with added sugars and ≥ 50 kcal per 8oz (236.5g) serving, including commercial or homemade beverages, soft drinks, energy drinks, fruit drinks, punch, lemonade, and frescas. This definition excludes 100% fruit and vegetable juices, non-caloric artificially sweetened drinks, and sweetened milk, tea, and coffee.” This definition was applied in a two-step process: initially, beverages with added sugars were identified by product name; and then the caloric threshold was applied to those products. During the time period of this evaluation, most SSBs contained about 100 kcal per 8oz, and thus this definition captured these. Added sugar content of foods is largely not available in dietary surveys due to methodological and analytical constraints (Eur J Clin Nutr 2015 <https://doi.org/10.1038/ejcn.2014.256>). In fact, the Global Dietary Database assessed added sugar as separate exposure, and it was only available in <45 surveys. Thus, our definition leveraging product name and caloric content to identify beverages with added sugar across the world ensures consistency in reporting. We have clarified these important points in our Methods section.

Changes to text (pg. 20; Line 432-437): “SSBs were defined based on its caloric content as any beverages with added sugars (as identified by product name) and ≥ 50 kcal per 8oz (236.5g) serving, including commercial or homemade beverages, soft drinks, energy drinks, fruit drinks, punch, lemonade, and frescas. This definition excluded 100% fruit and vegetable juices, non-caloric artificially sweetened drinks, and sweetened milk, tea, and coffee. All included surveys used this definition.”

• **Comment 3:** I think the very low estimates in Asia are due to the omission of teas. In country-reports show very different estimates of SSB intake when these are included. For example, if you look up country reports by the ministry of health or NGOs (Bangladesh heart foundation) you find a very different estimate of SSBs than what is presented in this study. It is not clear why authors did not include sweetened milk, tea, and coffee in their definition of SSBs. Perhaps it is due to the complexity of defining these in the data; however every effort should be made to do this, or else the main results of this study will be seriously biased. Many of these drinks are pre-sweetened and purchased ready to drink and should absolutely be included. Some will have sugar added during preparation or at the table; every effort should be made to include these, just

as you include other homemade drinks. Even those these could not be included in policies like SSB taxes, since they don't contain added sugar at the time of purchase, they are critical for estimating consumption, since they contain added sugar at the time of consumption.

Response: We appreciate your thoughtful comment. Given potential differential health impacts of beverages that include added sugar but have additional nutrients, such as calcium, vitamin D, protein in milk, and caffeine and polyphenols in coffee and tea, we did not include those in our definition of sugar-sweetened beverages. The definition we used included products with sufficient evidence of association with cardiometabolic disease. For instance, in our latest meta-analysis, we did not find sufficient evidence of association for fruit juice and coffee with cardiometabolic disease, while tea was found to reduce the risk of stroke (JAMA 2022 doi:[10.1001/jamanetworkopen.2021.46705](https://doi.org/10.1001/jamanetworkopen.2021.46705)). We agree that understanding and evaluating the potential health impacts of sweetened milk, as well as ready-to-drink (RTD) teas and coffee merits further investigation, but this is outside the scope of the present analysis that focuses on SSBs with established health harms. We have further clarified the rationale for not including sweetened milk, coffee, and tea in the revised Discussion section.

We also appreciate the point on country reports that may report different findings than ours – here understanding the data source and its definition is critical to the comparisons. For the example you noted for Bangladesh, we looked at the report by the Directorate General of Health Services, Ministry of Health and Family Welfare, and Government of the People's Republic of Bangladesh¹. We noticed that the findings are based on Euromonitor data which uses sales data, thus not accounting for actual intake and potential waste, and therefore not fully comparable to our estimates. Regardless, the Bangladesh report suggested that national SSB intakes of 0.2 8oz servings/week (2.4 liters/person/year) in 2004 and 0.5 8oz servings/week (5.6 liters/person/year) in 2018, which compares to our estimates of 0.3 8oz servings/week for 2005 and 0.2 8oz servings/week in 2018 (Supplementary Table 7, **pg. 51**). Of note, Euromonitor defines soft drinks as carbonates, fruit/vegetable juice, bottled water, functional drinks, concentrates, RTD tea, RTD coffee and Asian specialty drinks, thus also not including milk.²

Changes to text (pg. 14, Lines: 288-302): “We did not include sugar-sweetened milks, which are widely excluded from the definition of SSBs in policy and surveillance; nor home-sweetened teas or coffees (which often would have less than 50 kcal/serving). Additionally, global dietary surveys generally do not collect information on sweetened tea or coffee, and even sales data on ready-to-drink tea and coffee is limited. Therefore, by excluding these products we enhance comparability across countries and findings from other reports. Future surveillance of the intake of ready-to-drink (RDT) teas and coffee is needed given that this is a rapidly expanding market, particularly in Asia. Although fruit juices (100%) contain free sugars, observational analyses suggest no major associations of 100% fruit juice with key disease outcomes. Therefore, these were excluded in our definition of SSBs, which focuses on beverages with added sugars. The WHO recommends limiting the intake of added sugars to less than 10% of the total caloric intake, with additional benefits at intakes lower than 5%. Our findings on high intakes of SSBs and increases overtime across most regions, support the global actions aimed at decreasing the

¹ Noncommunicable Disease Control Programme (2021). Technical report on: Policy options for taxing sugar-sweetened beverages in Bangladesh. Directorate General of Health Services, Ministry of Health and Family Welfare, Government of Bangladesh. Available: <https://cdn.who.int/media/docs/default-source/searo/bangladesh/publications/technical-report-on-policy-options-for-taxing-sugar-sweetened-beverages-in-bangladesh-april-2021.pdf>

² <https://www.euromonitor.com/soft-drinks-in-the-us/report>

intake of SSBs. Moreover, highlight the need of targeting all sugar-containing beverages, including sweetened teas and milks, to prevent substitution with these beverages.”

Changes to text (pg. 14, Lines: 285-287): “Our findings might differ from estimates from sales data. However, the later do not account for nonconsumption due to waste nor estimate a population “mean” due to skewness of actual intakes.”

Changes to text (pg. 16, Lines: 326-328): “Particularly, there were limited surveys identified for South Asia (n=9) and Sub-Saharan Africa (n=22), which might have limited the accuracy of our estimates in those regions (Supplementary Table 4).”

• **Comment 4:** Similarly, consider including sweetened milks. Sweetened milks are controversial since dairy contributes some benefits; yet at the same time, their exempt status from SSB taxes has largely been the product of industry lobbying. The WHO recommends including sweetened milks in SSB definitions for policy purposes (<https://www.who.int/publications-detail-redirect/9789240056299>). At a minimum, I suggest a sensitivity analysis of your main findings including coffee, tea, and sweetened milks.

Response: As described above, the international collection of global dietary surveys generally did not collect information on sweetened milk consumption, and these had been widely excluded from SSB definitions in policy and surveillance. For example, in a meta-analysis looking at the association of SSB intakes and weight gain, the SSB definition for most studies did not include milk (AJCN 2023 <https://doi.org/10.1016/j.ajcnut.2022.11.008>; see summary table below). While this may be partly due to industry lobbying, it may also relate to the positive nutritional qualities of milk, compared with sugar-sweetened water (Adv Nutr 2023 [PMID: 36914032](https://pubmed.ncbi.nlm.nih.gov/36914032/)). We have added some discussion of the lack of these data in the discussion as shown in the previous comment. In addition to this, a cross-sectional study in Mexico found that, although caloric beverages were the major contributors to total caloric intake, milk intake was not one of the leading beverages among adults (JN 2014 [PMID: 24744311](https://pubmed.ncbi.nlm.nih.gov/24744311/)).

SSB definitions of 10 studies included in Metanalysis for adults: Nguyen et. al. AJCN. 2023. Sugar-sweetened beverage consumption and weight gain in children and adults: a systematic review and meta-analysis of prospective cohort studies and randomized controlled trials	
Study	SSB definition
B.J. Auerbach, A.J. Littman, J. Krieger, B.A. Young, J. Larson, L. Tinker, et al. Association of 100% fruit juice consumption and 3-year weight change among postmenopausal women in the in the Women's Health Initiative. Prev Med , 109 (2018), pp. 8-10	Sugar-sweetened beverages (regular soda and Tang ®, Kool-Aid ®, Hi-C ®, or other < 100% fruit juice drinks)
M.T. Barrio-Lopez, M.A. Martinez-Gonzalez, A. Fernandez-Montero, J.J. Beunza, I. Zazpe, M. Bes-Rastrollo. Prospective study of changes in sugar-sweetened beverage consumption and the incidence of the metabolic syndrome and its components: the SUN cohort. Br J Nutr , 110 (9) (2013), pp. 1722-1731	SSBs: sugar-sweetened carbonated colas and fruit-flavoured carbonated sugar soft drinks
M.E.C. Buso, E.M. Brouwer-Brolsma, N.D. Naomi, J.A. Harrold, J.C.G. Halford, A. Raben, et al. Dose-response and substitution analyzes of sweet beverage consumption and body weight in Dutch adults: the Lifelines Cohort Study Front Nutr , 9 (2022), Article 889042	SSB was defined as soda sugar drinks or lemonade (both carbonated and non-carbonated).
L. Chen, L.J. Appel, C. Loria, P.H. Lin, C.M. Champagne, P.J. Elmer, et al. Reduction in	SSBs (regular soft drinks, fruit drinks, fruit punch, or any other high-calorie beverage sweetened with sugar)

consumption of sugar-sweetened beverages is associated with weight loss: the PREMIER trial	
R. Gonzalez-Morales, F. Canto-Osorio, D. Stern, L.M. Sánchez-Romero, L. Torres-Ibarra, R. Hernández-López, et al. Soft drink intake is associated with weight gain, regardless of physical activity levels: the health workers cohort study Int J Behav Nutr Phys Act , 17 (1) (2020), p. 60	The FFQ included two items on the intake of soft drink (cola and flavored sodas)
C.G.Y. Lim, C. Whitton, S.A. Rebello, R.M. van Dam Diet quality and lower refined grain consumption are associated with less weight gain in a multi-ethnic Asian adult population. J Nutr , 151 (8) (2021), pp. 2372-2382	SSB: estimated using the soft drink variable
J. Ma, N.M. McKeown, S.J. Hwang, U. Hoffmann, P.F. Jacques, C.S. Fox. Sugar-sweetened beverage consumption is associated with change of visceral adipose tissue over 6 years of follow-up Circulation , 133 (4) (2016), pp. 370-377	To estimate SSB intake, we summed consumption of (1) caffeinated colas with sugar, (2) caffeine-free colas with sugar, (3) other carbonated beverages with sugar, and (4) fruit punches, lemonade, or other noncarbonated fruit drinks.
P. Mullie, P. Autier, M. Boniol, P. Boyle, B. Deforche, E. Mertens, et al. Assessment of sugar-sweetened beverage consumption and weight change: a prospective cohort study	We defined SSB as beverages containing added caloric sweeteners such as sucrose and/or high-fructose corn syrup.
A.C. Nooyens, T.L. Visscher, A.J. Schuit, C.T. van Rossum, W.M. Verschuren, W. van Mechelen, et al. Effects of retirement on lifestyle in relation to changes in weight and waist circumference in Dutch men: a prospective study Public Health Nutr , 8 (8) (2005), pp. 1266-1274	Sugared soft drinks
A.O. Odegaard, W.P. Koh, K. Arakawa, M.C. Yu, M.A.A. Pereira Soft drink and juice consumption and risk of physician-diagnosed incident type 2 diabetes: the Singapore Chinese health study Am J Epidemiol , 171 (6) (2010), pp. 701-708	Two different questions from the food frequency questionnaire specifically asked study subjects to report the intake frequency of 1) soft drinks such as Coca-Cola, and 2) other fruit and vegetable juices

• **Comment 5:** You could also consider including 100% juices. Though these do not contain added sugar, they do contain free sugars and are identified by the WHO as sugary drinks that should be limited or reduced (<https://www.who.int/publications/i/item/9789241549028>).

Response: Fruit juices (100%) were collected as a separate exposure and were not included as part of the definition of SSB. This is consistent with national policies and surveillance on SSBs, which exclude 100% juice. Moreover, contrary to SSBs, 100% fruit juices are not significantly associated with cardiometabolic disease (JAMA 2022

doi:[10.1001/jamanetworkopen.2021.46705](https://doi.org/10.1001/jamanetworkopen.2021.46705)). Based on the depth, detail, and complexity of our current paper, expanding this publication to include both SSBs and 100% fruit juices – which have different health and policy implications – would not be practical. Fruit juices (100%) are planned to be presented as part of future papers by the group. This has been further clarified in the manuscript, as mentioned in *changes to text* in comment #3 above.

• **Comment 6:** FFQs and the corresponding method to convert into servings most likely underestimate the consumption, which could have biased the results towards the null, particularly among high consumers. It would be helpful to include a table showing the % of studies that used different diet assessment methods as well as other relevant information on the diet data used (average sample size & range, average calories intake & range, etc.).

Response: We agree that repeated 24-hour recalls and food records are the gold standard for assessing mean nutritional intakes of populations. Yet, SSB intakes assessed by food frequency questionnaires (61% of our surveys) have been validated against 24-hour recalls. For instance, Vanderlee et al. (BMC 2018 DOI: [s12937-018-0380-8](https://doi.org/10.1186/s12937-018-0380-8)) conducted a validation of a 17-item Beverage Frequency Questionnaire vs. a 7-day food record and found statistically significant correlations in the volume reported for regular soda ($r=0.51$, $p\text{-value}<0.001$), fruit drinks with added sugars ($r=0.56$, $p\text{-value}<0.001$), flavored water with calories $r=0.95$, $p\text{-value}<0.001$) and sport drinks ($r=0.82$, $p\text{-value}<0.001$). Most of these also had non-significant p-values for the difference in volume analyzed using t-test. Supplementary Table 4 (pg. 21) displays the percentage of surveys by dietary method including food frequency questionnaire (61%), 24-hour recall (24%), household budget survey (12%), and demographic and health survey data (3%). The representativeness is also included (most surveys were nationally or subnationally representative) as well as the total sample of subjects included globally and for each region. In addition, we now show Supplementary Table 6 (pg. 23-50) which includes details on each survey included in the model for the estimation of SSB intakes.

Changes to text (pg.15, Lines: 314-317): “Our estimates are informed by individual-level dietary data collected from both 24-hour recalls (24% of surveys), a gold standard for assessing nutritional intake of populations, and food frequency questionnaires (61% of surveys), a validated approach for measuring SSB intakes (Supplementary Table 4).”

Changes to text (pg.19, Lines: 413-414): “Further details on survey characteristics in Supplementary Table 6.”

• **Comment 7:** Include prevalence of consumers. This result is essential. Particularly for the low-consuming countries, the servings per week are less informative. What % of people are even consuming these drinks once a week? You present data on the high consumers but I also want to know the "any" consumers.

Response: We agree the prevalence of consumers is an interesting separate question. Our data collection methods and model generate mean estimates at the stratum level (e.g., mean intake of SSBs among females 20-24 years old, from rural areas, from low-education level in the US) and not at the individual level. Thus, we lack the information to individually allocate participants as consumers. One of the strengths of the GDD is the dissemination of the original individual-level microdata in our website (<https://www.globaldietarydatabase.org/>) for all surveys providing permission. Thus, calculating the prevalence of consumers for individual nations and years with available surveys could be part of a separate analysis using the original microdata, although not part of the Bayesian modeling estimates and out of the scope of this current study.

• **Comment 8:** In general, the discussion was lacking in-depth integration and interpretation of results. Specific suggestions follow: The discussion section could have provided more context about differences among the regions in terms of the economy, nutrition and epidemiologic transition, availability, and accessibility of SSBs, and existent policies, similar to what was done with Sub-Saharan Africa. Also, results from the implementation of policies that could have reduced SSBs intake besides taxation, particularly in Latin America, could have been included, as well as potential consequences, such as product reformulation.

Response: We appreciate these detailed suggestions of each item that could improve our discussion. We have edited the Discussion section accordingly. Note that we removed “sales

restrictions in schools” from the policies previously mentioned given that this paper is focused on adults.

Changes to text (pg. 10-11, Lines: 202-224): “Increasing trends were more pronounced in specific subnational groups, and with varying patterns in these groups by world region. For example, increases from 1990 to 2018 were larger among youngest adults in Central/Eastern Europe and Central Asia, Middle East/ North Africa, South East and East Asia and Sub-Saharan Africa (Supplementary Table 17), while in other world regions, trends varied less by age. Trends were not notably different between male and female globally or regionally. Increases in intakes were higher in rural than in urban areas in Middle East/North Africa and Southeast and East Asia, but higher in urban than in rural areas in Sub-Saharan Africa and Central/Eastern Europe and Central Asia. By education level (a proxy of socioeconomic status), increases in SSB intakes from 1990 to 2018 were greater among the lowest educated adults in High-Income Countries, Middle East/North Africa, and Southeast and East Asia, but greater among the highest educated adults in South Asia and the middle-educated adults in Sub-Saharan Africa. Based on these trends, by 2018 highest overall SSB intakes were seen among the most educated adults in Sub-Saharan Africa and Latin America/Caribbean (Table 1).

The inverse national correlation between SDI and SSB intake in 2018 represents cross-country comparisons of national development, as opposed to within-nation socioeconomic status of individuals. The SDI findings highlight cross-national inequities in intakes, showing that higher national social and economic development is statistically significantly correlated with lower the SSB intakes. In comparison, within nations, socioeconomic status as measured by education has contrasting relationships with SSB intakes in different regions and nations, with higher educated adults often consuming more SSBs in most nations with lower SDI. These findings are in line with the ongoing nutrition and epidemiologic transition globally, disproportionately affecting the poorest nations.”

Changes to text (pg. 11-12, Lines: 227-237): “The World Health Organization (WHO) has widely recommended SSBs taxes as one of the main evidence-based policy measures to reduce intake of SSBs. Nevertheless, many of these efforts have been blunted by strong food industry opposition techniques including disqualification of research findings, biased industry-funded research, misleading summaries, marketing techniques, and false claims on the potential adverse social consequences such as massive job losses. SSB taxes have been implemented in 108 nations globally, covering 52% of the world’s population, but most of these policies were implemented or updated after 2017 (n=71, 66%) (Supplementary Figure 8), and thus their impact is mostly not captured in SSB intakes up to 2018. Future surveillance of SSB intakes globally is needed to determine the relative impact of these and other policies on SSB intakes (such as reformulation), both across nations and for subgroups within each nation.”

Changes to text (pg. 13, Lines: 265-272): “Most national measures to reduce SSB intakes have occurred relatively recently, limiting their impact during the period of our investigation. In Latin America, for example, regulations on the advertising of SSBs have been implemented in Brazil (2006), Mexico (2009), Chile (2016), and Peru (2013); evidence on the impact of these measures remains limited, particularly for adults. Most recently, front-of-package “black box” warning labels for certain nutrients, including added sugars, have been implemented in several countries in Latin America. In Chile, an observational study of beverage purchases found that from 2015 to 2017 the purchase volume of SSBs decreased by 22.8ml per capita (23.7%) after implementation of the labels.”

• **Comment 9:** In addition to it, the discussion could have provided a comparison between the magnitude of the results and previous studies of SSBs, particularly in high-income countries, as well as age-specific policies, mostly notably Chile’s marketing ban, although results clearly showed that intake was higher among younger age groups. Including an interpretation of the results in comparison to current recommendations for added sugars/SSBs would also have been helpful.

Response: We addressed the Chile’s marketing regulation and younger ages in our response to comment #8 above; and regarding sugar intake recommendations in response to comment #3 above.

We looked at national SSB intakes from the High-Income Countries region and Chile and we found that estimates were slightly lower than ours (see table below). However, most of these studies defined SSBs as ‘soft drinks’, which generally refers only to sodas or similar drinks, thus, potentially excluding other sources of SSBs such as fruit drinks (not 100%) and home-made beverages. Therefore, we believe our estimates are in agreement with the current literature, while taking into consideration other sources of SSBs. We included comparison between previous studies in the Discussion section.

Changes to text (pg. 14, Lines: 381-385): “A few national studies using mostly 24-hour recalls or food records found slightly lower estimates than GDD. However, most of these defined SSBs only as soft drinks, which generally refers only to sodas or similar beverages, thus excluding other SSBs such as energy drinks, fruit drinks (not 100%), and homemade sweetened beverages which were included in our estimates.”

Comparison of individual studies SSB mean national intakes and GDD estimate for the closest study year.

Country	year(s)	n	sex	age	Dietary assessment method	Study 8oz serving/week	GDD 8oz serving/week	GDD year	Reference
Germany	2005-2006	7093	Men	18-64	Diet history interview, food record, 24-h recall	6.32	3.2	2005	Walton et al. ¹
	2005-2006	8278	Women	18-64		2.48	2.3	2005	Walton et al. ¹
France	2014-2015	1773	both	18-64	24 h recall	2.43	2.8	2018	Walton et al. ¹
Italy	2005-2006	1068	Men	18-64	3-day record	0.99	1.8	2005	Walton et al. ¹
	2005-2006	1245	Women	18-64		0.68	1.4	2005	Walton et al. ¹
Spain	2014-2015	993	both	18-74	24 h recall	1.86	3.1	2018	Walton et al. ¹
UK	2016-2019	1082	both	19-64	4-day record	2.99	4.4	2018	Walton et al. ¹
Chile	2015-2017	2,383 households	both	all	National data on household food purchases	2.66	8.3	2018	Taillie et al. ²

2015-2017	2,383 households	both	all	0.45	8.3	2018	Taillie et al. ²
-----------	------------------	------	-----	------	-----	------	-----------------------------

¹ Walton J, Wittekind A. Soft Drink Intake in Europe—A Review of Data from Nationally Representative Food Consumption Surveys. *Nutrients*. 2023; 15(6):1368. <https://www.mdpi.com/2072-6643/15/6/1368>

²Taillie LS, Reyes M, Colchero MA, Popkin B, Corvalán C (2020) An evaluation of Chile’s Law of Food Labeling and Advertising on sugar-sweetened beverage purchases from 2015 to 2017: A before-and-after study. *PLOS Medicine* 17(2): e1003015. <https://doi.org/10.1371/journal.pmed.1003015>

• **Comment 10:** The discussion section did not mention sex-differences in SSBs intake, and how they could have been due to energy adjustment procedures. Also, those and the adult male equivalent (AME) did not take into consideration regional differences in body composition, which could have impacted the results. These and other potential sources of bias could have been discussed with more detail.

Response: We did not separately estimate energy intake in each region, but we used the data on energy intake directly reported in the dietary surveys to standardize intake levels. We report findings adjusted to 2000 kcal/d for adults age 20-74 years and 1700 kcal/d for adults age 75+ years. This reduces variation in SSB intakes due to factors that affect total energy intakes, such as regional and sex differences in body size, metabolic efficiency, and physical activity. The adult male equivalent (AME) method was only used for 15 (3.3%) household-level surveys to convert the household data to individual data based on energy requirements for a specific age and sex. We did not find large differences in energy-adjusted SSB intakes by sex. We have clarified the above issues in the Methods and Discussion sections.

Changes to text (pg. 8, Lines: 150): “Energy-adjusted SSB intakes and trends were generally similar in males versus females.”

Changes to text (pg. 20, Lines: 429-432): “The adult male equivalent (AME) method was used for 15 household-level surveys (3.3% of all surveys) to convert the household data to individual data based on energy requirements for a specific age and sex (Supplementary Methods 1).”

• **Comment 11:** How do your results compare with other, potentially more objective data (i.e., Euromonitor sales data)? Self-reported dietary data would be much more compelling in this case if at least it shows similar trends to what is reported in other sources.

Response: We respectfully disagree that sales data are more “objective,” as sales data do not account for food waste nor other sources of SSBs, such as homemade sugar-sweetened drinks which are common in many nations. We now mention this in our Discussion section.

Changes to text (pg. 14, Lines: 285-287): “Our findings might differ from estimates from sales data. However, the later do not account for nonconsumption due to waste nor estimate a population “mean” due to skewness of actual intakes.”

Minor comments

• **Comment 12:** Abstract: is it possible to include 95% CI?

Response: We added the 95% UI to the abstract.

• **Comment 13:** I'm not sure if reporting the mean global increase in SSB servings is that useful, because surely it disguises a great deal of heterogeneity across countries as well as consumption.

Could you present averages by region? Or, change in the prevalence of SSB consumers? I think this may be more telling.

Response: We believe the mean global increase is relevant, as well as regional and national changes, which are all presented, overall, and jointly stratified by population characteristics between 1990, 2005, and 2018 (pg. 7-9, Lines: 137-176). As mentioned in our response to comment #7, calculating the prevalence of consumers for individual nations and years with available surveys could be part of a separate analysis using the original microdata, although not part of the Bayesian modeling estimates and out of the scope of this current study.

• **Comment 14:** Background: implies SSB intake causes social inequities, whereas it is the reverse that is true.

Response: By relationship we mean that the two factors are related without implying causality or directionality. We have changed this to “association”.

• **Comment 15:** Background: If the framing is that nutrition leads to morbidity and mortality globally, I think you need to mention undernutrition as well as overnutrition, i.e., the double burden. SSBs fit into this picture also, given that they do not contribute nutritional benefits and yet, in many countries, have been promoted/ marketed to low-income populations, exacerbating the double burden.

Response: In the introduction we mention that in 2019 poor dietary habits, overweight/obesity and **undernutrition** contributed to diet-related disease burdens, and that most of these were due to cardiometabolic diseases including cardiovascular disease, type 2 diabetes, and cancer (pg. 3, Line: 32-33).

• **Comment 16:** Methods: I would like to have seen family income included in the stratifications, mostly as another source of inequality that urbanicity and education attainment alone might not have evidenced.

Response: We agree that family income could further inform inequities in intakes. During the methodological development of the GDD, global experts concluded that household income is reported with significant error and bias in different countries, due to, for example, hidden sources of income and tax evasion. Thus, the decision was made to assess education as a more stable, comparable measure of socioeconomic status. This has been clarified in the Methods supplement.

Changes to text (Supplementary Material pg.5): “We used education as a measure of socioeconomic status, as based on expert consultation this was felt to be a more stable and comparable measure than other variables which can be reported with significant error and underreporting in different countries (e.g., household income)”.

• **Comment 17:** Although the methods are robust, the Bayesian method only provide an estimate and did not indicate differences between subgroups. Please further clarify in the manuscript text that you are extrapolating trends and not making estimates based on observed data points.

Response: Our model estimates intakes for years in which data were available. This has been further clarified in our response to comment #1. We calculated differences between subgroups at the stratum-specific level, using all 4000 posterior estimates to account for the full spectrum of uncertainty. We clarified this in the Methods section.

Changes to text (pg. 22, Lines: 481-483): “Absolute changes and percentage changes in consumption between 1990, 2005, and 2018 were calculated at the stratum-specific level using all 4,000 posterior estimates to account for the full spectrum of uncertainty.”

• **Comment 18:** Did you include interaction terms in the model (for example, interaction between sex/education or education/urbanicity, and age/urbanicity)?

Response: We decided not to include interaction terms between various demographic variables to preserve model stability. Given the data typically available globally for a given dietary factor, adding interaction terms would create substantial complexity for the model and result in unstable estimates. This has been added to the limitations section.

Changes to text (pg. 16, Lines: 333-337): “Furthermore, assumptions relating to standardization of serving sizes, SSB definitions, energy adjustment, and household-level disaggregation, as well as of no interaction between sociodemographic variables in our model, could have impacted our estimates. We decided not to include interaction terms between various demographic variables to preserve model stability.”

• **Comment 19:** How was missing data handled? If you lacked dietary data on an age group in a country, for example, how did you include this? Consider conducting sensitivity analyses to determine if the imputation method changed results.

Response: We do not use an imputation model. Our Bayesian hierarchical model uses survey data and country-year specific covariates to generate SSB estimates for all countries, years, and all 264 strata. For countries in which data were not available, the mean intakes were informed by the country-year specific covariates, the regional means, and the within-region variance (Supplementary Methods **pg. 15**). The hierarchical structure assumed in our model allows us to borrow strength across units, a concept known as “partial pooling”, in which each country’s mean estimate borrows from the other countries’ data within the region, resulting in shrinkage of the country mean estimate towards the region mean. The less informative the data were for a particular country, the more pooling there is. When data were available for a country, but not for a particular stratum within that country the estimates were informed by the other components in the model as described in the Supplementary Methods (**pg. 12-13**). For age, the model uses region-specific splines to estimate the age trends as explained in Supplementary Methods (**pg. 14**). We evaluated multiple different estimation models, and the best model was selected using five-fold cross-validation (randomly omitting 20% of the raw survey data, run five times), with validity further assessed by comparing predicted vs. observed intakes, excluding implausible estimates (Supplementary Table 2, **pg. 8**), and by visual assessment of global and national mean intakes using heat maps. This is now explained more clearly in the Methods section.

Changes to text (pg. 21; Lines: 457-461): “We evaluated multiple different estimation models, and the best model was selected using five-fold cross-validation (randomly omitting 20% of the raw survey data, run five times) and validity was further assessed by comparing predicted vs. observed intakes, excluding implausible estimates (Supplementary Table 2), and by visual assessment of global and national mean intakes using heat maps.”

• **Comment 20:** It is not clear how the SDI was used- you controlled for this? It's not clear what "assessing trends according to SDI" means.

Response: We did not adjust for SDI. We assessed the correlation between national SSBs intakes and SDI for 1990, 2005, and 2018 to observe the association between these two variables

(Figure 7, pg. 44), and how this association changed over time. We have revised this sentence to make it clearer.

Changes to text (pg. 23, Lines: 487-489): “We also assessed the correlation between country-level SSB intakes and the corresponding national sociodemographic development index (SDI), and how these relationships changed over time between 1990, 2005, and 2018.”

• **Comment 21:** Why use means instead of medians or quantile regression? Was the distribution of SSB intake right skewed, as is typical of dietary data?

Response: We agree that intakes of most dietary factors including SSBs are generally right skewed. The shape of the usual intake distribution is not tested in our research question. The purpose of the model is to estimate mean intakes for various subgroups within every country-year based on reported mean intakes. Mean intakes are the common metric reported globally, thus, we aimed to be consistent when reporting our estimates. Of note, the GDD estimation model operates on the natural log scale, including the covariate data. Thus, the country-level means are distributed log-normally for a given world region. This is consistent with what we have observed in our data.

• **Comment 22:** Results: For looking at age, I think it is less interesting to compare age groups across countries or regions than to look at the old- young age difference within each country and then compare those. You've already told us that SSB intake is low in Asia relative to Lat Am, so presenting the findings that younger adults in Asia are also low is less interesting. Do they consume more than older adults?

Response: We present trends in SSBs intakes by age in 2018 in Figure 2 (pg. 37), and for years 1990 and 2005 in Supplementary Figure 4 (pg. 70). We edited the corresponding text in the results section to expand on the explanation of this findings.

Changes to text (pg. 5, Lines: 86-92): “By age, SSB intakes were higher at younger vs. older ages in all regions, though with varying absolute magnitudes of intakes and differences by region (Table 1; Figure 2). For instance, in Latin America/Caribbean, where intakes were the highest compared to all regions, adults 20-24 years had a mean SSB intake of 11.1 servings/week while adults 85+ had a mean intake of 3.9 servings/week. In contrast, in South Asia, where intakes were the lowest across all regions, intakes were 1.0 servings/week and 0.4 servings/week in adults age 20-24 and 85+ years respectively.”

• **Comment 23:** Results: Useful to present urbanicity results separately before by education

Response: Thank you for the suggestion. We edited Figure 4 (pg., 39; former figure S3) and added Supplementary Figure 5 (pg.71). We also included relevant text in the results section.

Changes to text (pg. 7, Lines: 123-126): “Globally, intakes were higher in urban vs. rural areas. However, regionally this was only the case in Sub-Saharan Africa and South Asia; whereas the intake was inverse in Middle East/North Africa; and intakes almost the same between urban and rural areas in all other regions (Figure 4a).”

• **Comment 24:** Results: Line 133- this is the first time I'm seeing you use the "high income country" categorization which seems heterogeneous to how you've presented the results so far. Also- where does Europe & Australia fit in here? High-income countries could be a useful way to distinguish global trends but be sure to define & use consistently throughout .

Response: Our model estimates SSB intakes for seven world regions. One of these regions has been labeled as “High-income countries”, although it does not contain all countries classified as high-income by the World Bank. Our team and others (e.g., Global Burden of Disease study) previously used this world regions classification, and we aimed to be consistent. More explanation on the world regions and the countries within each region is available in Supplementary Methods (pg.11) and Supplementary Table 3 (pg. 20).

• **Comment 25:** Line 156: typo (Sub-Sharan)

Response: This has been fixed.

• **Comment 26:** Supp page 52: typo (Global Fund Research Program)

Response: We appreciate the careful review of the figure/table legends. The latest information for this figure is now derived from a different source and this has been replaced accordingly.

Thanks for the opportunity to review this paper. I look forward to reading future versions and seeing the eventual publication.

Response: Thank you for your thorough review and detailed feedback for strengthening this publication.

Reviewer #2 (Remarks to the Author):

The present study provides novel evidence on the consumption of sugar-sweetened beverage consumption for 185 countries and provides novel insights into time trends as well as variation in consumption levels by education and urbanicity. The authors provide a very comprehensive explanation of the complex methodological approach used by them. The results are timely and of substantial value for policy makers worldwide.

Response: Thank you for your kind comment and for agreeing on the potential value of our work for policy makers worldwide.

Below are some comments that this reviewer would like to see addressed before publication of the paper:

Major:

Comment 1: Abstract: Compared to the wealth of data provided in the article the abstract is quite vague. This applies specifically to the conclusion which should be more specific. The authors are encouraged to shorten the background sentence and to incorporate some of the novel findings (e.g. that SSB consumption emerges as a public health problem in sub-Saharan Africa).

Response: We agree that the abstract was lacking this information. We edited the findings and conclusion sections of the abstract as suggested.

Changes to text (pg. 1-2; Lines 20-28): “In 2018, mean global SSB intake was 2.7 (8oz=248g) servings/week (95% UI 2.5-2.9), (range: 0.7 (0.5-1.1) in South Asia to 7.8 (7.1-8.6) in Latin America/Caribbean). Intakes were generally higher among male vs. female, younger vs. older, more vs. less educated, and urban vs. rural adults. Variations by education and urbanicity were

largest in Sub-Saharan Africa. Between 1990 and 2018, global SSB intakes increased by +0.37 (+0.29, +0.47), with the largest increase in Sub-Saharan Africa. These findings inform intervention, surveillance, and policy actions worldwide, highlighting the growing problem of SSBs for public health in Sub-Saharan Africa.”

Comment 2: US-centric presentation: Given that this is a global report and that – ask acknowledged by the authors – most countries report intakes in g (or ml) it is surprising that all data are still given servings based on ounces. At the very least should the authors include the g/ml equivalent in the abstract. Note that in Table S6 the conversion is explained as 8oz=237 g, whereas in most other footnotes 8oz=248 g is given.

Response: A good point. We have added the conversion to the abstract and the first time mentioned in the results. We have also corrected the conversion in Supplementary Table 7 (pg. 51; former table S6) to 8oz=248g.

Comment 3: Similarly, the authors should not only refer to “added sugar”, but also to “free sugar”, i.e. the WHO definition which is used by many countries around the globe.

Response: Free sugar includes natural sugar in juices. The evidence suggests that drinks with natural free sugar (100% juice) do not have similar health harms as SSBs (JAMA 2022 doi:[10.1001/jamanetworkopen.2021.46705](https://doi.org/10.1001/jamanetworkopen.2021.46705)). We added explanatory text in the discussion.

Changes to text (pg. 14, Lines: 294-297): “Although fruit juices (100%) contain free sugars, observational analyses suggest no major associations of 100% fruit juice with key disease outcomes. Therefore, these were excluded in our definition of SSBs, which focuses on beverages with added sugar.”

Comment 4: Country-specific sources of data: while this reviewer acknowledges that a complex model was used to estimate the consumption in the respective countries and that this approach allowed to report intake levels for 185 countries despite the fact that at least one report of survey data was only available for 118 countries it would be very informative for both the reader and local policy makers if the authors provided an overview of the data sources used for the estimates in each country.

Response: Thank you for the suggestion. We added a table in the supplement with the characteristics for each survey included in the modeling for SSBs (Supplementary Table 6 pg. 23-50) and we refer to it in the Methods section.

Comment 5: Differences by education vs relation of SSB intake to SDI: At first glance it appears contradictory that intake levels should be higher among persons with a higher educational level in most regions whilst on the other hand Figure 5 suggests an inverse correlation between national SDI and SSB intake for 2018. This is presumably due to the former being an observation based on individual data while the latter is a correlation on national level. Yet, this reviewer feels that this should be explained in the discussion together with the implications arising from this for policy makers.

Response: Excellent point. Our education level category is based on individual-level data, while the SDI is based on country-level data. We clarified this in the discussion section.

Changes to text (pg. 11; Lines: 216-222): “The inverse national correlation between SDI and SSB intake in 2018 represents cross-country comparisons of national development, as opposed to within-nation socioeconomic status of individuals. The SDI findings highlight cross-national

inequities in intakes, showing that higher national social and economic development is statistically significantly correlated with lower the SSB intakes. In comparison, within nations, socioeconomic status as measured by education has contrasting relationships with SSB intakes in different regions and nations, with higher educated adults often consuming more SSBs in most nations with lower SDI.”

Comment 6: Reference to tax policies: While it is correct that many Latin American countries still have no taxes implemented this is however the region (together with Europe) with the broadest coverage of tax policies. This could be acknowledged in the context of the discussion of the results for this region.

Response: We agree that the Latin American/Caribbean region and Europe today are regions with more common SSB taxes. However, most of these taxes were implemented or updated after 2017 (Supplementary Figure 8, **pg. 80**), thus their impact in our modeling of SSB intakes up to 2018 is not relevant. We added explanatory text to the discussion.

Changes to text (pg. 11; Lines 231-234): “SSB taxes have been implemented in 108 nations globally, covering 52% of the world’s population, but most of these policies were implemented or updated after 2017 (n=71, 66%) (Supplementary Figure 8), and thus their impact is mostly not captured in SSB intakes up to 2018.”

Minor:

• **Comment 7:** Page 15, lines 23-26: this is difficult to understand. Please, explain more explicitly what is hidden and why this in line with the inverse correlation seen in 2018.

Response: This has been addressed in comment #5 above.

• **Comment 8:** Please, mention in the introduction that this report focuses on findings for adults aged 20 years onwards.

Response: We now mention that this study focuses on adults age 20+ years in the introduction.

• **Comment 9:** Please, explain why out of 1,248 surveys only 451 report data on SSB

Response: The GDD identified, harmonized, standardize 1,248 surveys containing one or more of the 53 dietary factors if interest. Many of these surveys reported intakes only on a few foods, and 451 provided information on SSBs and were used for modeling SSB intakes.

Of note, we had previously reported 1,248 as the main number of surveys in GDD. However, some of those surveys were not included in the model. We now report 1,226 surveys, which were those that were used to model the different dietary factors. We updated the manuscript accordingly. The number of surveys used for modeling SSB intakes remained unchanged (451 surveys).

• **Comment 10:** Line 139 ff, page 20: please explain that energy adjustment was made via the residual method whenever possible. This is explained in the supplemental material, yet readers not familiar with energy adjustment may interpret reference to 2000 kcal or 1700 kcal as “energy adjustment”.

Response: We now mention that we adjusted for total energy through the residual method in the methods section.

Changes to text (pg. 20; Lines 426-428): "...and adjusting for total energy through the residual method and using age-specific energy intakes to reduce measurement error and account for regional differences in body size, metabolic efficiency, and physical activity."

• **Comment 11:** Lines 151 ff, page 21: This reviewer struggles to understand why data from children and adolescents were needed to estimate intake levels among adults. A short explanatory would be much appreciated.

Response: Although this paper is focused on adults, the purpose of our model is to create estimates not only for adults, but for all age groups for all country years. Thus, each estimate is informed by all the data we have. Modeling all age groups together allows to use the full set of available data to inform our estimates, including age patterns, relationships between predictors and SSB intakes, and covariates for bias control (e.g., dietary assessment methods). Thus, by modeling all data together we get a more informed model.

Reviewer #3 (Remarks to the Author):

The paper is of high quality but I have some comments/queries as described below.

Response: Thank you for your comment.

I have carried out a statistical review:

Comment 1: The methods section refers to the posterior predictive. Should this be the posterior? That would be the more natural estimate to report.

Response: Thank you for this comment. Yes, this should be "posterior", not "posterior predictive". We have corrected the main manuscript and supplement accordingly. (**pg. 21, Line: 454; and Supplementary Methods pg. 15**).

Comment 2: As part of the cross-validation, was any check made of the validity of the UIs? If not, this should be done.

Response: The UIs represent statistical uncertainty, and not an estimate of any "true" measure, therefore, to our knowledge there are no validity checks for this.

Comment 3: In 'Statistical Analysis', I think that 'posterior predictions' should be replaced by 'posterior estimates' (consistent with point 1).

Response: We agree that 'posterior estimates' is more accurate than 'posterior predictions'. We have updated the main manuscript and supplementary material appropriately (**pg. 22, Lines: 473 & 482; and Supplementary Methods pg. 16**).

Other minor comments:

Comment 4: I think the metric equivalent of the 8 oz serving should be given in the findings and main text, particularly as the paper is intended for an international policymaking audience.

Response: We added the conversion to the abstract and the first time it is mentioned in the results.

Comment 5: In the introduction, is it correct to say that the poor dietary habits etc definitely caused these deaths? Would text such as 'are associated with' or 'contributed to' be more accurate?

Response: We changed the text accordingly.

Changes to text (pg. 3; Line 32-33): “In 2019, poor dietary habits, overweight/obesity, and undernutrition contributed to 14%, 10%, and 5% of deaths...”

Comment 6: In Figure 2, I think joining the points is misleading unless it really is intended that readers should interpolate between them. I'd suggest removing the lines and making the points and CIs larger.

Response: Thank you for your feedback on this figure. The range of the uncertainty intervals (UIs) is small for most regions, thus by making the points larger we would hide information on the UIs (e.g., see example figure below). Therefore, keeping the points at the current size is the best solution to preserve display of the UIs, and the connecting lines provide a solution to help visually see the differences within regions as well as across regions.

Comment 7: In the Figure 4 caption, "the filled bars represent the absolute change..." seems to refer to (B) only. Should the text be modified here?

Response: Yes, we corrected accordingly.

Changes to text (pg. 40; Legend figure 5 (former Figure 4)): “The filled bars represent the mean intake in (a) and the absolute change in the mean intake in (b). The error bars represent the 95% UI of these respective values.”

Comment 8: A further limitation that should be added in that section is that the data are self-reported. Is there anything in the literature on how well people taking the surveys recall their SSB consumption?

Response: We acknowledge that all dietary data collection methods have limitations. For example, self-report relies on memory, which could lead to under or over-reporting. Repeated 24-hour recalls and food records are the gold standard for assessing mean nutritional intakes of populations. Yet, SSB intakes assessed by food frequency questionnaires (61% of our surveys) have been validated against 24-hour recalls. For instance, Vanderlee et al. (BMC 2018 DOI: [s12937-018-0380-8](https://doi.org/10.1186/s12937-018-0380-8)) conducted a validation of a 17-item Beverage Frequency Questionnaire vs. a 7-day food record and found statistically significant correlations in the volume reported for regular soda ($r=0.51$, $p\text{-value}<0.001$), fruit drinks with added sugars ($r=0.56$, $p\text{-value}<0.001$), flavored water with calories ($r=0.95$, $p\text{-value}<0.001$) and sport drinks ($r=0.82$, $p\text{-value}<0.001$). Most of these also had non-significant p-values for the difference in volume analyzed using t-test. Supplementary Table 4 (pg. 21) displays the percentage of surveys by dietary method including food frequency questionnaire (61%), 24-hour recall (24%), household budget survey (12%), and demographic and health survey data (3%).

Changes to text (pg. 15; Lines: 314-317): “Our estimates are informed by individual-level dietary data collected from both 24-hour recalls (24% of surveys), a gold standard for assessing nutritional intake of populations, and food frequency questionnaires (61% of surveys), a validated approach for measuring SSB intakes (Supplementary Table 4).”

Changes to text (pg. 16; Lines: 328-331): “All types of dietary assessments include some error, whether from individual-level surveys, national food availability estimates, or other sources. Our model’s incorporation of multiple types and sources of dietary assessments provide the best available estimates of global diets, as well as the uncertainty of these estimates.”

Comment 9: This is more an editorial point, but should it be stated somewhere which country the Consejo Nacional de Ciencia y Tecnología is from?

Response: Consejo Nacional de Ciencia y Tecnología is a Mexican institution. We now added this information (pgs. 29-30; Lines: 726 and 743).

The supplementary material should be proof read to the standard of the paper. There are quite a few typos, including some in mathematical formulae. Some are mentioned below.

Specific comments:

Comment 10: Typo in caption for Fig S1 ('standardize')

Response: We corrected the title for Supplementary Figure 1 (pg. 19).

Comment 11: On p10, could the use of entry and exit points be clarified? It's not clear what p is here.

Response: Entry and exit points refer to the threshold of the correlation p-value for a given covariate to be included or excluded in the stepwise regression. In our final model we only included covariates with p-value <0.299 (entry point). When the p-value of a given covariate was

>0.30 (exit point), either originally or after adding other covariates, it was excluded from the model. Note that we had erroneously typed exit point as $p < 0.30$, but it should be $p > 0.30$. This has been corrected.

Changes to text (Supplementary Methods pg. 9): “Each of the covariates identified in the correlation stage (maximum 10 covariates) and the four PCA components were then included in a stepwise regression to test for inclusion in GDD models, with an entry point p-value < 0.299 and an exit point of > 0.30 during the stepwise process.”

Comment 12: p11. typo for 'descriptions' at the bottom of the page

Response: This has been corrected (Supplementary Methods **pg. 12**).

Comment 13: p13. typo for 'standardized' at top of page.

Response: This has been corrected (Supplementary Methods **pg. 13**).

Comment 14: p14. various subscripts are not subscripted in the text, and in the description of the age model: 'Zh'

Response: This was a typo. The correct subscript is Z_h , which corresponds to “midpoint age of stratum h in study”, as explained in Supplementary Methods **pg. 12**. This has been corrected. We have also reviewed/edited the supplement to ensure all subscripts are explained (Supplementary Methods **pgs. 12-17**).

Comment 15: p14. I think this should be 'Estimates' rather than 'Predictions'. It looks like you're (correctly) reporting the posterior rather than the posterior predictive. I'm not sure why you wouldn't have a posterior estimate of the estimate for a country with no data as this would be informed by the other data and the covariates; similarly with superregions with no data.

Response: We agree that ‘estimates’ is more accurate than ‘predictions’ and we have updated the main manuscript and supplement accordingly. In our supplement text we mention “For countries with no survey data, we did not have posterior distribution for a_j (country intercept).” This means that a_j is not a parameter in the model when country j doesn't exist in our data. Thus, for these countries we report the distribution of a_j^* , which propagates the within-region uncertainty into the estimates for the country. Given that countries with no data are essentially unobserved data points, we were referring to the estimates for these countries as “predictive distributions”. However, we now understand that such terminology might be erroneously confused with “posterior predictive distributions” which is not what we calculated. We changed our language to “posterior distributions” (**pg. 21, Line 454; Supplementary Methods pgs. 15**)

Comment 16: For the varying slopes model, it would be helpful to include the time-varying component more explicitly in the mathematical representation by adding time-based subscripts.

Is it that you have $\mu_{i,t} = \alpha_{\text{country}[i, \text{reference year}] + \beta_{\text{country}[i]} * \text{FAO}_t + M_{\text{method}[i]}$?

Response: There was no time varying component in the varying slopes model. The core idea is that "time" is captured by the underlying time variation of FAO/Genus covariates, as well as the second Bayesian model capturing the covariate variables that change with time. This has been clarified in the manuscript.

Changes to text (pg. 21-22; Line 461-470): “A second Bayesian model was used to strengthen time trend estimates for dietary factors with corresponding food or nutrient availability data from

FAO Food Balance Sheets or the Global Expanded Nutrient Supply. The model incorporated country-level intercepts and slopes from these covariates, along with their correlation estimated across countries. No time-component was formally included in the model; rather, time was captured by the underlying time variation in the model covariates. This model is commonly referred to as a varying slopes model structure and leverages two-dimensional partial pooling between intercepts and slopes to regularize all parameters and minimize overfitting risk. The final presented results are a combination of these two Bayesian models, as detailed in Supplementary Methods 3.”

REVIEWER COMMENTS

Reviewer #1 (Remarks to the Author):

The authors did a nice job of addressing reviewer comments. I appreciate the additional context on the modeling used, which was quite rigorous. As noted before, I do think this paper could provide an important advancement in our understanding of global SSB intake.

However, 2 critical points remain in my eyes:

1. The 50 calorie cutpoint per 8 oz for SSBs is arbitrary and not in line with global recommendations. There is not, to my knowledge, any set of recommendations or policies that uses this definition for SSBs. More importantly, given that the vast majority of beverages with added sugar get their calories primarily from sugar, the 50 calorie cutoff is likely way too high. Under this definition, a drink could contain 10 grams of sugar per 8 oz (or approx 20 grams of sugar per 100 ml) and not be considered an SSB. This is a much higher sugar level than permitted by any policy definition on SSBs. This means that the results presented in this paper are likely a vast underestimation of global SSB intake. Since it doesn't seem like authors can include actual sugar amounts, I urge authors consider an alternate definition with a lower calorie threshold.

2. I still think that the lack of inclusion of teas & coffees is majorly problematic. These are SSBs and arguably the most important ones in Asia; this is also where the market is growing the fastest. Even if this data does not include information on those products, authors could use other datasources (Euromonitor) to look at global trends in sales of these beverages to, at a minimum, contextualize the potential gaps in the data they present. I see a major risk in failing to do this, which is that the results could be interpreted such that Asia does not have a problem with SSBs, when we know from other reports that consumption of these products is growing rapidly there.

Reviewer #2 (Remarks to the Author):

Thank you for thoroughly addressing all points raised.

Reviewer #3 (Remarks to the Author):

The authors have responded to all of my major points. I have a series of remaining minor points, most of which are very minor:

1. Line 18: typo (missing space).
2. Line 41: the sentence 'intensive marketing to traditionally marginalized populations' should be referenced.
3. Line 63: typo after 'Caribbean'.
4. Line 84: Germany has 0.88 greater difference in males vs females, rather than the 1+ quoted. Should be US, Mexico and France.
5. Line 144: there is a discrepancy with Supplementary Table 7, with -1.22 in the text and -1.23 in the table.
6. Line 191: typo (missing space).
7. Line 207: should be 'males and females'.
8. Line 219: typo ('the' at the end of the line should be deleted).
9. Line 222: 'consuming more SSBs in most nations with lower SDI' should be referenced.
10. Lines 267-268: suggest swapping Chile and Peru so the list is in chronological order.
11. Line 285: 'latter' rather than 'later'.
12. Line 300: should be 'over time'.
13. Line 301: should be 'Moreover, this highlights the need to target' or something similar.
14. Line 307: suggest 'Bayesian hierarchical models'.
15. Line 424: should be 'including averaging'.
16. Line 479: should be 'in FoodData Central'.
17. Line 516: should be 'developed using R'.
18. Line 517: should be 'desired strata categories'.
19. Figure 4 caption: should be 'The values below the bars correspond...'
20. References: some minor formatting issues such as the bold in ref 31 and '&mdash' in ref 37.

REVIEWER COMMENTS

Reviewer #1 (Remarks to the Author):

The authors did a nice job of addressing reviewer comments. I appreciate the additional context on the modeling used, which was quite rigorous. As noted before, I do think this paper could provide an important advancement in our understanding of global SSB intake.

Response: We appreciate your thoughtful feedback.

However, 2 critical points remain in my eyes:

Comment 1: The 50 calorie cutpoint per 8 oz for SSBs is arbitrary and not in line with global recommendations. There is not, to my knowledge, any set of recommendations or policies that uses this definition for SSBs. More importantly, given that the vast majority of beverages with added sugar get their calories primarily from sugar, the 50 calorie cutoff is likely way too high. Under this definition, a drink could contain 10 grams of sugar per 8 oz (or approx 20 grams of sugar per 100 ml) and not be considered an SSB. This is a much higher sugar level than permitted by any policy definition on SSBs. This means that the results presented in this paper are likely a vast underestimation of global SSB intake. Since it doesn't seem like authors can include actual sugar amounts, I urge authors consider an alternate definition with a lower calorie threshold.

Response: According to Malik and Hu (Nature 2022 <https://doi.org/10.1038/s41574-021-00627-6>) there is currently no universally agreed-upon definition on SSBs. The calorie content per 8 oz serving typically ranges between 84 to 104 calories for regular sodas, 106 to 154 calories for energy drinks, 114 to 134 for juice drinks, and 114 to 216 for aguas frescas, as reported in the USDA database. These values are generally at least twice the 50-calorie cutoff point that we have set for our definition. As a result, our definition encompasses all regular soft drinks and even some with less sugar than average. However, it does not include homemade sweetened tea, which typically contains 15 to 30 calories per 8 oz serving (equivalent to 1 to 2 packs of sugar). It is worth noting that homemade tea is traditionally prepared unsweetened in Asia, where it is most commonly consumed. Furthermore, until recently, there were very few soft drinks available in the market with calorie contents ranging between 30 to 50 calories per 8 oz serving, which includes beverages sweetened with a combination of caloric and non-caloric sweeteners. Hence, our definition might not capture a few of these drinks, but they still represent a relatively small portion of the overall soft drink market.

The data on SSB intake was collected using the 50-calorie cutoff point definition, as provided by the data owners. In most cases the sugar content information was not included. Therefore, we currently lack the necessary data to calculate the SSBs intake based on a different calorie threshold.

Changes to the manuscript (Lines: 301-306): "Our definition may have missed beverages with lower sugar content, but the vast majority were captured. For instance, the calorie content per 8 oz serving is typically between 84 to 104 calories for regular sodas, 106 to 154 calories for energy drinks, 114 to 134 for juice drinks, and 114 to 216 for aguas frescas. As a result, our definition encompasses all usual SSBs and even some with less sugar than average."

Comment 2: I still think that the lack of inclusion of teas & coffees is majorly problematic. These are SSBs and arguably the most important ones in Asia; this is also where the market is growing the fastest. Even if this data does not include information on those products, authors could use other datasources (Euromonitor) to look at global trends in sales of these beverages to, at a minimum, contextualize the potential gaps in the data they present. I see a major risk in failing to do this, which is that the results could be interpreted such that Asia does not have a problem with SSBs, when we know from other reports that consumption of these products is growing rapidly there.

Response: We examined the sales data trends of ready-to-drink (RTD) teas in Asia, utilizing the Statista Markets Insight tool. The data revealed that the sales of RTD teas, which include packaged tea beverages in liquid form and ready for consumption, either prepared at home or away from home, were relatively modest in 2018. On average, consumption stood at 0.33 8oz servings per week*, indicating slight growth compared to 2014. As a result, we conclude that the inclusion of RTD teas in our estimates would not exert a significant impact. We have incorporated this information into our discussion.

Changes to text (Lines: 294-296): “Furthermore, sales data suggest that ready-to-drink (RTD) tea consumption was relatively modest in Asia in 2018 (0.33 8-oz servings/week), showing slight growth in comparison to 2014. Thus, inclusion of these teas would not substantially alter our results. However, the Asian market for RTD teas, along with RTD coffees, is expanding, and future surveillance and monitoring is needed to keep pace with this evolving category.”

* 4.11 lts per year = 139.0 oz per year = 17.4 8oz servings per year = 0.33 8oz servings per week

Source: Ready-to-Drink (RTD) Tea - Asia. (n.d.). Retrieved May 30, 2023, from <https://www.statista.com/outlook/cmo/non-alcoholic-drinks/ready-to-drink-rtd-coffee-tea/ready-to-drink-rtd-tea/asia?currency=USD&locale=en>

Reviewer #2 (Remarks to the Author):

Thank you for thoroughly addressing all points raised.

Reviewer #3 (Remarks to the Author):

The authors have responded to all of my major points. I have a series of remaining minor points, most of which are very minor:

Response: Thank you for your thorough review of our paper.

1. Line 18: typo (missing space).

Response: Space added

2. Line 41: the sentence 'intensive marketing to traditionally marginalized populations' should be referenced.

Response: The reference was misplaced. This has been fixed

3. Line 63: typo after 'Caribbean'.

Response: '.' added

4. Line 84: Germany has 0.88 greater difference in males vs females, rather than the 1+ quoted. Should be US, Mexico, and France.

Response: We deleted 'Germany' from that sentence.

5. Line 144: there is a discrepancy with Supplementary Table 7, with -1.22 in the text and -1.23 in the table.

Response: Corrected to "-1.23" in the text as well

6. Line 191: typo (missing space).

Response: Space added

7. Line 207: should be 'males and females'.

Response: Corrected

8. Line 219: typo ('the' at the end of the line should be deleted).

Response: Corrected

9. Line 222: 'consuming more SSBs in most nations with lower SDI' should be referenced.

Response: This sentence is based on our findings (Figure 4). We edited to make it clearer.

Changes to text (Lines: 220-223): "In comparison, within nations, socioeconomic status as measured by education has contrasting relationships with SSB intakes in different regions, with higher educated adults often consuming more SSBs in regions mostly composed by countries with lower SDI such as in Latin America/Caribbean, South Asia, and Sub-Saharan Africa (Figure 4)."

10. Lines 267-268: suggest swapping Chile and Peru so the list is in chronological order.

Response: Corrected

11. Line 285: 'latter' rather than 'later'.

Response: Corrected

12. Line 300: should be 'over time'.

Response: Corrected

13. Line 301: should be 'Moreover, this highlights the need to target' or something similar.

Response: Corrected

14. Line 307: suggest 'Bayesian hierarchical models'.

Response: Corrected

15. Line 424: should be 'including averaging'.

Response: Corrected

16. Line 479: should be 'in FoodData Central'.

Response: Corrected

17. Line 516: should be 'developed using R'.

Response: Corrected

18. Line 517: should be 'desired strata categories'.

Response: Corrected

19. Figure 4 caption: should be 'The values below the bars correspond...'.

Response: Corrected

20. References: some minor formatting issues such as the bold in ref 31 and '—' in ref 37.

Response: Corrected

REVIEWERS' COMMENTS

Reviewer #1 (Remarks to the Author):

The authors have adequately addressed my remaining concerns. I think this manuscript will provide a strong contribution to the literature on global SSBs and I recommend it for publication.

1

2 **REVIEWERS' COMMENTS**

3

4 Reviewer #1 (Remarks to the Author):

5

6 **Comment 1:** The authors have adequately addressed my remaining concerns. I think this
7 manuscript will provide a strong contribution to the literature on global SSBs and I recommend
8 it for publication.

9

10 **Response:** We are delighted to hear your positive response to our work and we appreciate all
11 your thoughtful feedback.